# DC-W2S: Dual-Consensus Weak-to-Strong Training for Reliable Process Reward Modeling in Biological Reasoning

Chi-Min Chan [1 2 * †]   Ehsan Hajiramezanali [2 *]   Xiner Li [2]   Edward De Brouwer [2]   Carl Edwards [2]   Wei Xue [1]
Sirui Han [1]   Yike Guo [1]   Gabriele Scalia [2]

## Abstract

In scientific reasoning tasks, the veracity of the reasoning process is as critical as the final outcome. While Process Reward Models (PRMs) offer a solution to the coarse-grained supervision problems inherent in Outcome Reward Models (ORMs), their deployment is hindered by the prohibitive cost of obtaining expert-verified step-wise labels. This paper addresses the challenge of training reliable PRMs using abundant but noisy "weak" supervision. We argue that existing Weak-to-Strong Generalization (W2SG) theories lack prescriptive guidelines for selecting high-quality training signals from noisy data. To bridge this gap, we introduce the Dual-Consensus Weak-to-Strong (DC-W2S) framework. By intersecting Self-Consensus (SC) metrics among weak supervisors with Neighborhood-Consensus (NC) metrics in the embedding space, we stratify supervision signals into distinct reliability regimes. We then employ a curriculum of instance-level balanced sampling and label-level reliability-aware masking to guide the training process. We demonstrate that DC-W2S enables the training of robust PRMs for complex reasoning without exhaustive expert annotation, proving that strategic data curation is more effective than indiscriminate training on large-scale noisy datasets.

## 1. Introduction

Within recent years, scientific discovery has emerged as a critical application of AI advances (Zhang et al., 2023; Jumper et al., 2021). Biological applications are particularly challenging, requiring holistic integration of heterogeneous knowledge across scales (Xu et al., 2023; Moor et al., 2023). Large Language Models (LLMs) with "reasoning" capabilities have shown transformative potential for this integration, using natural language as a unifying medium for biological evidence (Istrate et al., 2025; Wang et al., 2025).

A dominant paradigm for aligning LLMs with domain-specific tasks is Reinforcement Learning using Verifiable Reward (RLVR), which often relies on Outcome Reward Models (ORMs) to optimize for correct final answers (Ziegler et al., 2019; Cobbe et al., 2021; Ouyang et al., 2022; Bai et al., 2022). However, this sparse reward risks inadvertently validating reasoning trajectories that are flawed, illogical, or factually incorrect, so long as they coincidentally arrive at the correct final output (Zelikman et al., 2022; Creswell et al., 2022; Lyu et al., 2023; Turpin et al., 2023). This failure mode is particularly pernicious in scientific domains, including biology and healthcare. For example, predicting downstream effects of perturbations requires multi-step reasoning through target engagement and pathway interactions across heterogeneous biological contexts (e.g., cell types, tissues, disease states). A model that "hallucinates" a plausible pathway, yet guesses the right answer, is arguably more dangerous than one that is transparently wrong, as it can mislead researchers and result in the catastrophic waste of experimental time and resources. Thus, ensuring the veracity of the reasoning process is vital, yet challenging.

A direct solution to this limitation is the shift from outcome-based to process-based supervision. Unlike ORM, a Process Reward Model (PRM; Lightman et al., 2023; Wang et al., 2024) evaluates each intermediate step of a Chain-of-Thought (CoT; Wei et al., 2022), providing a dense, fine-grained reward signal. This granular feedback enables precise localization of errors, teaching the model how to reason correctly rather than merely the correct answer. This capability is critical for moving beyond simple answer-retrieval and toward genuine, verifiable mechanistic insights in biology.

Training PRMs at scale, however, requires step-level supervision that is often impractical and prohibitively expensive to obtain from biological domain experts. Scalable automated alternatives, such as Monte Carlo (MC) estimations (Wang et al., 2024; Luo et al., 2024) and LLM-as-a-

---

[1]The Hong Kong University of Science and Technology, Hong Kong [2]Genentech, Inc., South San Francisco, CA, USA. Correspondence to: Ehsan Hajiramezanali <hajiramm@gene.com>.

*Proceedings of the $43^{rd}$ International Conference on Machine Learning*, Seoul, South Korea. PMLR 306, 2026. Copyright 2026 by the author(s).

judge (Zheng et al., 2023), alleviate the manual-labeling burden, but generate inherently noisy weak labels that lack expert-verified ground truth (Zhang et al., 2025b). Naively training on these weak annotations risks the "garbage in, garbage out" problem, where the resulting PRM merely learns to imitate the systemic errors and biases of its automated teachers.

Under this context, we ask: **"How can we train a strong, reliable PRM by leveraging only these abundant, but imperfect, weak label sources?"** Existing theoretical frameworks on Weak-to-Strong (W2S) generalization offer a partial explanation for why sufficiently robust students can learn from weak supervision under favorable data structure, e.g., when the weak supervisor's error sets are sparsely distributed and surrounded by correctly labeled neighbors (Burns et al., 2023; Zhou et al., 2025; Lang et al., 2024). Yet existing theory is primarily post hoc and descriptive rather than prescriptive. As a result, it provides limited actionable mechanisms to actively curate training data in the absence of ground truth, and its applicability to complex biological reasoning remains unestablished.

To address these challenges, we propose the Dual-Consensus Weak-to-Strong (DC-W2S) training framework. Our core insight is that not all weak labels contribute equally to the generalization of a strong student; while some provide robust learning signals, others introduce detrimental noise. We facilitate a "teacher-centric" curation by evaluating step-level annotations via two orthogonal metrics: (1) *Self-Consensus* (SC), which measures the agreement across heterogeneous weak supervisors (e.g., Monte Carlo estimation and LLM-as-a-judge), and (2) *Neighborhood-Consensus* (NC), which quantifies label consistency within the step's neighborhood (defined semantically or biologically). By intersecting these metrics, we stratify the supervision space into four distinct reliability regimes as P1 (SC & NC), P2 (SC & ¬NC), P3 (¬ SC & NC) and P4 (¬ SC & ¬ NC). Building on this stratification, we propose a two-level anchored training strategy that improves efficiency through (i) a distribution-aware sampling curriculum over reliability regimes and (ii) reliability-aware loss masking that suppresses gradients from redundant or ambiguous supervision. Together, these components provide a practical foundation for reducing annotation burden in future biological reasoning tasks.

To the best of our knowledge, this is the first systematic study of W2S generalization for PRMs in biological reasoning under step-wise weak supervision. In particular, we focus on single-cell perturbation prediction, a key task for understanding biological systems due to the quantity of available data (Zhang et al., 2025a) and the transferability of insights to other biological scales (Ma et al., 2021). Our main contributions are: (1) We construct a large-scale

perturbation reasoning trajectory dataset with multi-source weak step annotations, which we will release to support future research. (2) We introduce DC-W2S, a dual-consensus framework that stratifies weak step labels by intersecting Self-Consensus with Neighborhood-Consensus, enabling an anchored training strategy that selectively exploits reliable supervision rather than training indiscriminately on noisy labels. (3) We provide theoretical analyses of PRM learning under aggregated weak step supervision, deriving error bounds under soft robust expansion assumptions. (4) On biological perturbation reasoning, our experiments show that DC-W2S improves PRM robustness and label efficiency, achieving competitive performance with fewer weakly labeled steps and demonstrating positive transfer across tasks/settings.

## 2. Related Works

**LLM for Biological Reasoning.** LLMs have evolved from static knowledge bases into reasoning engines for computational biology. Recent frameworks such as BioReason (Fallahpour et al., 2025) and ChatNT (de Almeida et al., 2025) integrate genomic encoders with LLM backbones; rbio1 (Istrate et al., 2025) introduces a reasoning model trained with biological world models; and TxGemma (Wang et al., 2025) adopts agentic workflows to verify biological hypotheses. For our critical task of predicting cellular responses to genetic perturbations, while embedding-centric models such as scGPT (Cui et al., 2024), GenePT (Chen & Zou, 2024), and GEARS (Roohani et al., 2024) excel in endpoint accuracy, they often function as "black boxes". In contrast, scientific inquiries require a mechanistic understanding of causal gene influences, which highlights the distinct advantage of LLMs (Wu et al., 2025). Our work shifts focus from outcome-based regression to process-oriented reasoning, leveraging LLMs to generate interpretable, multi-step traces grounded in verifiable biological mechanisms.

**Test-Time Scaling and Process Supervision.** Performance in complex reasoning can be significantly enhanced by increasing inference-time compute through parallel decoding or sequential refinement (Snell et al., 2024; Muennighoff et al., 2025). However, applying these scaling laws to biology remains challenging. Current RL paradigms rely on outcome-based feedback (Guo et al., 2025), susceptible to "reasoning hallucinations" in which models reach correct answers through flawed intermediate logic. In high-stakes scientific discovery, such hallucinations can be prohibitively costly, misdirecting hypotheses and wasting substantial wet-lab efforts. While PRMs (Lightman et al., 2023) provide step-wise feedback to mitigate this, the scarcity of human expert annotations has led to a reliance on automated yet noisy labeling techniques like Monte Carlo estimation or LLM-as-a-judge (Zhang et al., 2025b; Wang et al., 2024).

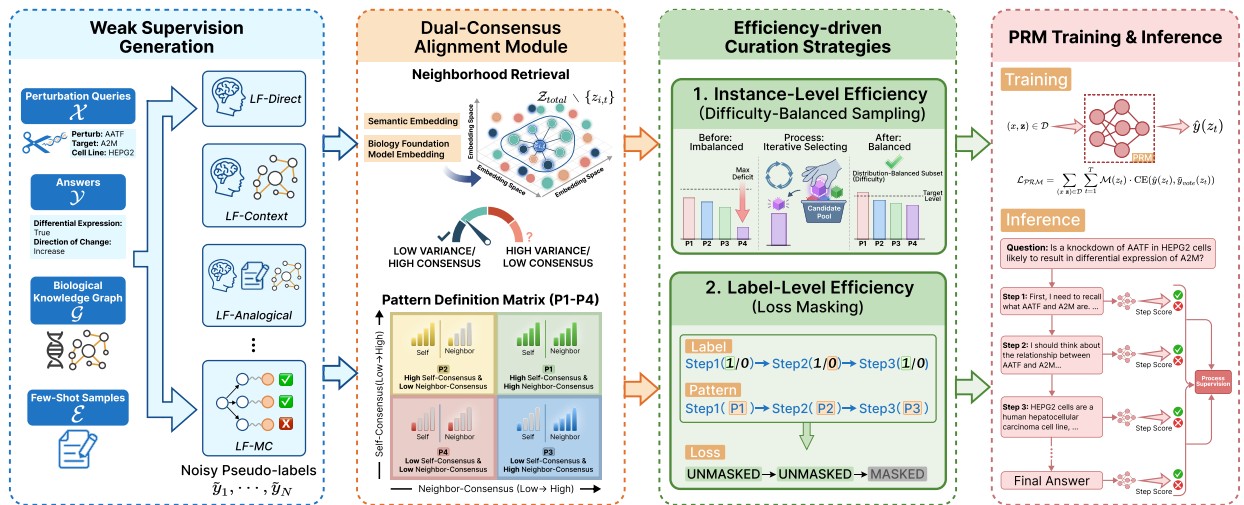

*Figure 1.* Dual-Consensus weak-to-strong supervision framework for efficient PRM training in biological reasoning, where expert step-level verification is costly or unavailable.

Thus, we introduce an anchored training algorithm designed to distill robust supervisory signals from these cost-effective but weak labels.

**Weak-to-Strong Generalization.** The study of learning from imperfect supervision generally uses label aggregation or supervision transfer. Traditional probabilistic approaches have focused on label aggregation through de-noising conflictive signals (Ratner et al., 2017; Dawid & Skene, 1979). The emerging paradigm of Weak-to-Strong Generalization (Burns et al., 2023; Xue et al., 2025) adopts naive majority voting to isolate the effects of training dynamics, demonstrating that strong student models can outperform their weak supervisors by leveraging superior internal representations. Drawing on insights from data pruning and coreset selection (Paul et al., 2021; Lang et al., 2022; Hu et al., 2024), our work investigates a novel aspect of supervision transfer to maximally elicit generalization, i.e., the geometry between training examples and decision boundary. We propose an anchored selection mechanism to identify high-value subsets within noisy datasets, optimizing the elicitation of strong performance from imperfect biological supervision.

## 3. Methodology

In this section, we propose Dual-Consensus Weak-to-Strong (**DC-W2S**), a framework for training a student PRM from noisy weak supervision. Figure 1 summarizes the pipeline: we (1) synthesize reasoning trajectories with context-aware prompting and obtain step-wise weak labels from heterogeneous supervisors (e.g., LLM judges and MC rollouts) and aggregate them; (2) stratify these labels via the Dual-Consensus mechanism into four label patterns, and (3) propose an anchored training strategy to enable effective W2S generalization at both instance-level and label-level.

### 3.1. Preliminaries: Process Reward Modeling

We use biological reasoning as a concrete instantiation of our framework, since supervision is often limited to question–answer pairs $(x, y_{\text{final}})$, and obtaining reliable labels for intermediate reasoning steps is particularly challenging in this domain. Given a question $x$ (e.g., *"Does ABCF1 knockdown in HepG2 affect ARCN1 expression?"*), a policy model generates $\mathbf{z} = (z_1, \dots, z_T)$ and a final predicted answer $\hat{y}_{\text{final}}$ (represented as the last step). We refer to each trajectory-level training example as an *instance* $(x, \mathbf{z}, y_{\text{final}})$, and to *labels* as step-wise supervision signals for individual steps $z_t$. A PRM $r_\theta$ assigns a score to each partial trajectory, producing step-wise rewards $s_t = r_\theta(x, z_{1:t}) \in [0, 1]$, and we use the shorthand $r_\theta(z_t) := r_\theta(x, z_{1:t})$. Our goal is to train $r_\theta$ using only noisy weak step supervision $\tilde{y}(z_t)$.

### 3.2. Weak Supervision Generation

Given the impracticality of collecting expert-curated labels for millions of reasoning steps, we rely on a scalable automated annotation strategy that can efficiently produce large volumes of supervision at minimal cost.

#### 3.2.1. SYNTHESIS OF REASONING TRAJECTORIES

In order to train a PRM, we first synthesize Chain-of-Thought (CoT) trajectories for the PerturbQA training set (Wu et al., 2025) using a Context-Augmented Generation strategy to ensure trajectory diversity and quality. For each query $x$, we retrieve relevant knowledge graph context (e.g., GO terms and pathway interactions) and a small set of similar training examples. We then sample a reasoning trajectory $\mathbf{z}$ (with final prediction $\hat{y}_{\text{final}}$) from a weak generator $\pi_{\text{gen}}$ (Qwen3-4B). This produces $\mathcal{D}_{\text{traj}} = \{(x^{(i)}, \mathbf{z}^{(i)}, y^{(i)})\}_{i=1}^{N}$ with 351k trajectories in total.

### 3.2.2. MULTI-SOURCE WEAK ANNOTATION

To assign a pseudo-label $\tilde{y}_t \in \{0, 1\}$ to each step $z_t$ without expert annotation, we collect weak supervision from two distinct classes of labeling functions (LFs) and aggregate their outputs into a single binary label per step.

**LLM-as-a-judge labeling.** We deploy an LLM-as-a-judge to evaluate each step's correctness. To mitigate prompt- and context-specific bias, we obtain labels from three complementary perspectives:

$$\begin{aligned}
\text{LF-Context}(z_t) &= \text{LLM}(z_{1:t}, \mathcal{G}, y_{\text{final}}), \\
\text{LF-Analogical}(z_t) &= \text{LLM}(z_{1:t}, (\mathcal{G}, \mathcal{E}), y_{\text{final}}), \\
\text{LF-Direct}(z_t) &= \text{LLM}(z_{1:t}, \varnothing, y_{\text{final}}),
\end{aligned}$$

where each function returns a binary judgment in $\{\text{Correct}, \text{Incorrect}\}$ (mapped to $\{1, 0\}$). Here, $\mathcal{G}$ denotes the full KG context, $\mathcal{E}$ is a set of few-shot examples, and $y_{\text{final}}$ is the ground truth final answer.

**Monte Carlo rollout labeling.** Complementing LLM-as-a-judge supervision, we use Monte Carlo (MC) rollouts to estimate whether a partial trajectory can be completed to the ground truth answer. We instantiate a set of MC labeling functions, one per rollout model $j \in \{1, \ldots, J\}$. Following Math-Shepherd (Wang et al., 2024), for each step $z_t$ we sample $K$ continuations from a completion policy and compute the success rate:

$$\text{LF-MC}_j(z_t) = \frac{1}{K} \sum_{k=1}^{K} \mathbf{1}\{\text{rollout}_k(z_t) \models y_{\text{final}}\}. \quad (1)$$

We binarize this score with threshold $\tau = 0.5$ to obtain a step label in $\{0, 1\}$. This label reflects the likelihood that continuing the reasoning from the current step will eventually reach the correct answer.

**Aggregation.** After collecting weak labels from multiple sources (LLM-based judge labeling functions and MC-based labeling functions), we derive the final step-level label via majority voting, i.e. $\tilde{y}_{\text{agg}}(z_t)$.

### 3.3. Dual-Consensus Weak-to-Strong Training Framework

Notably, the weak step labels collected in Section 3.2 are noisy and lack expert verification. We therefore estimate the *reliability* of each step label using two complementary signals: *Self-Consensus* (SC), measuring agreement across weak supervisors, and *Neighborhood-Consensus* (NC), measuring whether the step lies in a neighborhood that is consistently judged by the supervisors. We use these scores to stratify supervision for anchored training.

### 3.3.1. TEACHER-SELF-CONSENSUS (TSC)

TSC measures agreement among the $M$ heterogeneous LFs for a step $z_t$. Let $\ell_m(z_t) \in \{0, 1\}$ denote the binary label produced by LF $m$, and let $\text{Var}(\cdot)$ denote the empirical variance across labels. We define TSC as the normalized label concentration:

$$\text{TSC}(z_t) = 1 - 4 * \text{Var}(\{\ell_m(z_t)\}_{m=1}^{M}). \quad (2)$$

A high $\text{TSC}(z_t)$ implies that diverse LFs agree on the step's quality, indicating a signal that is robust to individual annotator biases. We call steps with $\text{TSC}(z_t) > \tau_{\text{sc}}$ *self-consensus* (self-reliable).

### 3.3.2. TEACHER-NEIGHBORHOOD-CONSENSUS (TNC)

TSC is a pointwise reliability estimate. We therefore define TNC to capture whether a step lies in a locally *unambiguous* region of the reasoning (embedding) space. Under our *reliability smoothness* assumption, reliable steps lie on a high-confidence manifold where heterogeneous weak supervisors exhibit low disagreement.

We first establish the local geometric structure in the semantic space. Using a pre-trained encoder $E_{sem}$ (e.g., Sentence-Transformers), we map each reasoning step $z_t$ to a dense vector. For a target step $z_t$, we retrieve its top-$K$ ($K = 20$) nearest neighbors $\mathcal{N}(z_t)$ from the trajectory bank based on cosine similarity. The TNC score is then defined as the expected reliability of the local neighborhood:

$$\text{TNC}(z_t) = \frac{1}{|\mathcal{N}(z_t)|} \sum_{z' \in \mathcal{N}(z_t)} \text{TSC}(z'). \quad (3)$$

A high $\text{TNC}(z_t)$ indicates that $z_t$ lies in a region where weak supervisors are consistently confident, and we mark steps with $\text{TNC}(z_t) > \tau_{\text{nc}}$ as *neighbor-consensus* (neighbor-reliable).

### 3.4. Biological Manifold Refinement for Neighborhood Construction

Semantic similarity alone can retrieve steps that are linguistically similar but biologically unrelated. To enforce biological coherence, we associate each step $z_t$ (via its originating query) with a perturbation gene $g_p$ and a target gene $g_t$, and define a biological context embedding $b(z_t) = [\phi(g_p); \phi(g_t)]$, where $\phi$ is a pretrained biological foundation model encoder over genes (e.g., ESM (Lin et al., 2023) or CellProfiler (Funk et al., 2022)). We then restrict candidate neighbors to steps whose biological contexts are similar, $\mathcal{C}(z_t) = \{z' : \text{sim}(b(z_t), b(z')) \geq \delta\}$, and finally compute $\mathcal{N}(z_t)$ as the semantic $k$-NN of $z_t$ within $\mathcal{C}(z_t)$. This refinement ensures that reliability smoothness is computed over a manifold that is both semantically aligned and biologically coherent.

## 3.5. Anchored Training Strategy

Based on the intersection of TSC and TNC, we stratify all training steps into four distinct reliability regimes (Figure 1): P1 (high SC & high NC), P2 (high SC & low NC), P3 (low SC & high NC) and P4 (low SC & low NC).

By default, we set $\tau_{sc} = \tau_{nc} = 0.5$. This is an unsupervised default requiring no validation-set tuning: since TSC $\in [0, 1]$ after variance normalization, $\tau_{sc} = 0.5$ on the normalized scale corresponds to majority agreement among labeling functions; $\tau_{nc} = 0.5$ similarly indicates that the local neighborhood exhibits above-median consensus.

We utilize this stratification through a two-level anchored training strategy.

### 3.5.1. INSTANCE-LEVEL STRATEGY: DISTRIBUTION-BALANCED SAMPLING

We observe that raw data distributions are often skewed: easy samples are dominated by P1 steps with trivial agreement, while others are overwhelmed by P4 noise. Training primarily on either extreme hinder W2S generalization. As a solution, we propose a distribution-balanced sampling curriculum. It iteratively selects an instance to add to the final subset. In each iteration, it calculates the current reliability pattern distribution across all previously selected instances. Afterwards, the new instance chosen is the one that has highest-density of the pattern that has largest deficit (the pattern farthest below the target 25% uniform distribution). This process continues until the target subset size is reached. The process is shown in Algorithm 1.

### 3.5.2. LABEL-LEVEL STRATEGY: RELIABILITY-AWARE LOSS MASKING

In addition to instance-level strategy, we also apply a masking mechanism to the training objective. We define the loss function as:

$$\mathcal{L}_{\mathcal{PRM}} = \sum_{(x,\mathbf{z})\in\mathcal{D}} \sum_{t=1}^{T} \mathcal{M}(z_t) \cdot \text{CE}(r_\theta(z_t), \tilde{y}_{agg}(z_t)), \quad (4)$$

where $\mathcal{M}(z_t) \in \{0, 1\}$ is the masking indicator. Our core hypothesis is that not all step-wise supervision is equally valuable; thus, the reliability patterns (P1–P4) potentially offer an interpretable basis for targeted supervision. Accordingly, we use pattern-specific masking to selectively include or suppress labels from particular regimes, enabling controlled ablations that quantify each pattern's contribution to performance and generalization (Section 4.2.4).

## 3.6. Theoretical Analysis

Here, we bound the ground truth step-label error of $r_\theta$ over the distribution of reasoning steps using terms involving the weak step-label error of $r_\theta$, i.e., $\text{err}_\tau(r_\theta, \tilde{y}_{agg})$, together with neighborhood expansion and robustness parameters. This bound provides a theoretical justification for why a reliability-aware loss with aggregated weak annotations can yield W2S generalization at the step level. Building on Lang et al. (2024), we introduce calibrated *soft* reliability weights $p(z_t) \in [0, 1]$ derived from our dual-consensus signals (SC/NC), and quantify "good" versus "bad" supervision via reliability-weighted masses. This soft formulation allows different regions of the reasoning-step distribution (e.g., with distinct reliability patterns) to contribute differently to the guarantee, providing a more faithful characterization of our training curriculum. Full proofs and extensions are deferred to the Appendix E.

**Setup.** For each step, the PRM outputs $r_\theta(z_t) \in [0, 1]$ and we consider $f_\tau(z_t) = \mathbf{1}\{r_\theta(z_t) \geq \tau\}$. Let $y(z_t)$ and $\tilde{y}_{agg}(z_t)$ be the latent and aggregated weak step labels, respectively. Define $\text{err}_\tau(r_\theta, y) = \Pr(f_\tau(z_t) \neq y(z_t))$ and $\text{err}_\tau(r_\theta, \tilde{y}_{agg}) = \Pr(f_\tau(z_t) \neq \tilde{y}_{agg}(z_t))$.

**Soft reliability masses.** Based on teacher consensus, let $p(z_t) \in [0, 1]$ denote the reliability of step $z_t$, interpreted as the probability that the aggregated weak-teacher label is correct.

**Assumption 3.1** (Calibration). For all steps $z_t$, $\Pr(\tilde{y}_{agg}(z_t) = y(z_t) \mid z_t) = p(z_t)$.

For any measurable set of steps $U$, define the reliability-weighted masses

$$\begin{aligned}
\mu_{\text{good}}(U) &= \mathbb{E}[p(z_t) \mid z_t \in U]\Pr(z_t \in U), \\
\mu_{\text{bad}}(U) &= \mathbb{E}[1 - p(z_t) \mid z_t \in U]\Pr(z_t \in U). \quad (5)
\end{aligned}$$

We also define the effective weak-label noise level $\alpha = \mathbb{E}[1 - p(z_t)]$.

**Definition 3.2** ($\eta$-robust). Fix a neighborhood operator $\mathcal{N}(\cdot)$ and a thresholded PRM decision rule $f_\tau$. For $\eta \in [0, 1]$, define the $\eta$-robust set as

$$R_\eta(f_\tau) := \left\{ z_t : \Pr_{z' \sim \mathcal{D}|\mathcal{N}(z_t)} \left[ f_\tau(z') \neq f_\tau(z_t) \right] \leq \eta \right\}.$$

**Definition 3.3** (Soft robust expansion). We consider that $(\mathcal{D}, \mathcal{N})$ satisfies $(c, q, \eta)$-soft robust expansion (w.r.t. $\mu_{\text{good}}, \mu_{\text{bad}}$) if for every measurable $U \subseteq R_\eta(f_\tau)$ with $\mu_{\text{good}}(U) \geq q$, $\mu_{\text{bad}}(\mathcal{N}(U)) \geq c\,\mu_{\text{good}}(U)$.

**Assumption 3.4** (Neighborhood label consistency). Let $\mathcal{N}(\cdot)$ denote the neighborhood operator induced by the embedding model. For every step $z_t \in R_\eta(f_\tau)$ and every neighbor $z' \in \mathcal{N}(z_t)$, the latent labels are locally consistent, i.e., $y(z') = y(z_t)$.

**Theorem 3.5** (Soft weak-label correction for PRM). *Assume Assumption 3.1 holds and that $(\mathcal{D}, \mathcal{N})$ satisfies $(c, q, \eta)$-soft robust expansion. Let $\bar{\rho}_\eta := \Pr(z_t \notin R_\eta(f_\tau))$*

and $c' := \frac{c}{(1-\alpha)+c\alpha}$. If

$$\Pr\Big(f_\tau(z_t) \neq \tilde{y}_{\mathrm{agg}}(z_t) \ \lor \ z_t \notin R_\eta(f_\tau)\Big) \ \leq \ 1 - q - \alpha,$$

then for any $\tau$ such that $1 - 2c'\alpha > 0$,

$$\mathrm{err}_\tau(r_\theta, y) \ \leq \ \frac{\mathrm{err}_\tau(r_\theta, \tilde{y}_{\mathrm{agg}}) \ - \ \alpha(2c' - 1) \ + \ 2c'\alpha\,\bar{\rho}_\eta}{1 - 2c'\alpha}.$$

*Remark* 3.6 (Informal: effect of miscalibration). If calibration is imperfect, we define the residual $\Delta(z_t) := \Pr(\tilde{y}_{\mathrm{agg}}(z_t) = y(z_t) \mid z_t) - p(z_t)$ and assume $\mathbb{E}[|\Delta(z_t)|] \leq \bar{\delta}$. Then Theorem 3.5 continues to hold up to an additional additive degradation of order $O\Big(\frac{\bar{\delta}}{1-2c'\alpha}\Big)$.

*Remark* 3.7 (Informal: connection to BoN). Our downstream objective is BoN trajectory selection using an aggregate PRM score $R_\theta(\pi) = \mathrm{Agg}_t(\{r_\theta(z_t)\})$. While weak-label correction need not be perfect at every step, reducing the step-level error rate improves the fidelity of these aggregated trajectory scores. As a result, tighter bounds on $\mathrm{err}_\tau(r_\theta, y)$ translate into tighter control of ranking noise, which increases the probability that BoN ranks higher-quality trajectories above lower-quality ones under a mild margin/separation condition.

## 4. Experiments

### 4.1. Experimental Setup

**Benchmarks.** We evaluate our framework on two biological reasoning benchmarks to assess both in-distribution performance and out-of-distribution (OOD) generalization. Our primary testbed is PERTURBQA (Wu et al., 2025), a gene perturbation dataset covering four cell lines. **For PRM training and evaluation**, we construct $\mathcal{D}_{\mathrm{train}}$ using K562, HepG2, and Jurkat cell lines, and hold out RPE1 entirely as an OOD test set, measuring generalization in an unseen cellular context. **For the SFT baseline**, we fine-tune using all four cell lines (including RPE1), and therefore report SFT results as in-distribution under this split. We report the F1 score as the primary metric; other metrics are listed in Appendix D. To assess cross-task transfer, we further evaluate checkpoints trained solely on PERTURBQA on BIOREASON dataset (Fallahpour et al., 2025), which comprises tasks that link genetic variants to pathogenicity. Dataset details are provided in Appendix B.

**Weak supervision and neighborhood construction.** We use the synthesized PerturbQA trajectories (Section 3.2.1) and obtain step-wise weak labels from heterogeneous supervisors: an LLM judge (Qwen3-32B) queried under three prompt context (*Context*, *Analogical*, *Direct*), and multiple MC rollout teachers instantiated with Qwen3-{1.7B,4B,7B}. We aggregate all weak labels by majority vote. For neighborhood consensus, we embed each

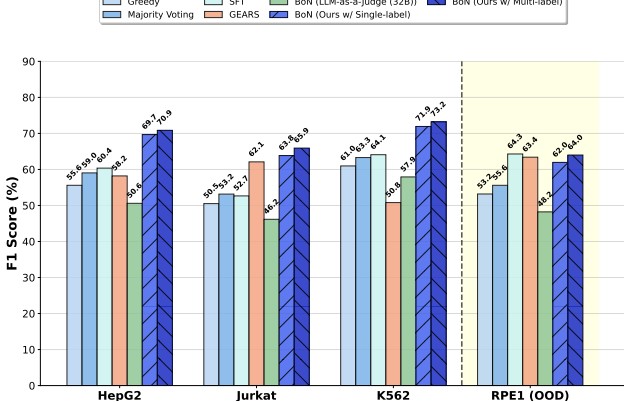

*Figure 2.* **Performance comparison across four cell lines. BoN (Ours w/ Full Set (Multi-Label))** achieves the highest average F1 scores, outperforming single-label and SFT baselines by effectively aggregating multiple weak supervisory signals. **Note:** RPE1 is OOD for PRM; SFT & GEARS is trained with RPE1.

step using all-MiniLM-L6-v2 (384-d) and perform cosine $k$-NN search with FAISS ($k = 20$). We also evaluate a biology-refined variant that filters candidate neighbors by biological similarity using precomputed gene embeddings from Littman et al. (2025) (concatenated perturbation/target gene embeddings) before semantic $k$-NN.

**PRM Training and Evaluation Protocol.** We initialize our PRM from Qwen3-4B, modifying the architecture by replacing the language modeling head with a scalar regression head, and train it with a binary cross-entropy objective against aggregated weak supervision. Additional implementation details are provided in Appendix A.

To evaluate its performance, we adopt a **Best-of-N** ($N = 8$) sampling protocol. Specifically, a fixed policy model (e.g. Qwen3-4B and RBIO1 (Istrate et al., 2025)) generates candidate reasoning trajectories using diverse decoding parameters (e.g. temperature $T = 0.7$), which are then scored by the PRM. The trajectory with the highest aggregated reward is selected as the final answer, and correctness is verified against the ground truth. We compare our PRMs against several baselines, including VersaPRM (Zeng et al., 2025), PRM800K (Lightman et al., 2023), and an LLM-as-a-judge baseline (Qwen3-32B), which is used to generate the training labels for our PRMs.

### 4.2. Main Results

Due to space constraints, we present representative results in the main text, primarily highlighting OOD performance. Full results—including IID evaluations, per-cell-line breakdowns, per-task performance, and ablations over embedding choices—are reported in Appendix D. Across these settings, the same conclusions hold.

#### 4.2.1. EFFECTIVENESS OF MULTI-SOURCE WEAK LABELS AGGREGATION

We first evaluate the effectiveness of the proposed multi-source weak supervision aggregation strategy across four representative cell lines (HepG2, Jurkat, K562, and RPE1). As shown in Figure 2, the Best-of-$N$ (BoN; $N = 8$) performance using multi-source supervision consistently outperforms training with a single weak source (*LF-Analogical* only), yielding an average F1 of **68.5%** versus **66.9%**. Moreover, our approach substantially exceeds the SFT[1] baseline (greedy decoding) of **60.4%** average F1, which directly optimizes the underlying policy model on all cell lines.

Beyond LLM-based methods, we further compare against GEARS (Roohani et al., 2024), a state-of-the-art bioinformatics foundation model that operates directly on numerical gene expression profiles. As shown in Figure 2, our method (Base Policy + PRM) consistently outperforms GEARS across all four cell lines, achieving absolute gains of +12.7, +3.8, +22.4, and +0.6 F1 on HepG2, Jurkat, K562, and RPE1, respectively. Notably, this comparison is conservative: GEARS is trained with access to RPE1 data, whereas our PRM treats RPE1 as strictly out-of-distribution and never observes it during training. These results indicate that discretizing high-dimensional numerical biological data into natural language does not incur a loss in predictive precision, while additionally offering interpretable step-by-step rationales that purpose-built numerical models cannot provide.

Above all, the results collectively suggest that aggregating feedback from diverse weak supervisors yields a more reliable step-wise training signal than any single weak source, translating into improved downstream policy performance.

#### 4.2.2. DIRECT EVALUATION OF STEP-LEVEL REASONING VALIDITY

One of the critical motivations of process-supervision is ensuring the validity of intermediate biological reasoning. To directly validate this, we conducted both targeted human expert annotations and larger-scale automated proxy evaluations.

**Human Expert Annotation.** A key concern is whether the PRM merely predicts final-answer correctness rather than tracking the scientific validity of each intermediate step. To test this, we selected 50 challenging questions whose sampled reasoning traces ($N$=8) exhibited diverse behaviors (i.e., high variance in both PRM scores and final-answer

---

[1]RPE1 is OOD only for PRM training (trained on K562/HepG2/Jurkat). The SFT & GEARS baseline is trained on all four cell lines (including RPE1), so the RPE1 comparison is conservative for our method.

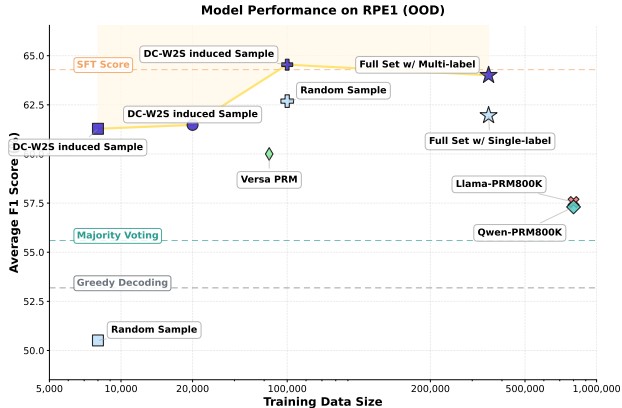

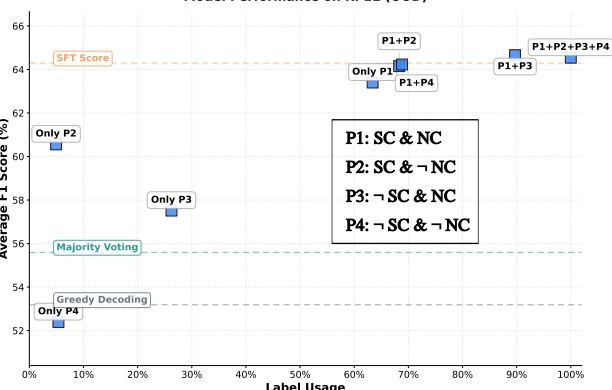

*Figure 3.* **Benchmarking Performance and Efficiency of BoN (Ours w/ CellProfiler). Top** (instance-level) and **bottom** (label-level) efficiency analyses show that we can achieve comparable performance while using fewer training instances and fewer weak step labels, thereby suggesting the effectiveness of the DC-W2S.

correctness). Rather than exhaustively annotating all traces, we applied stratified random selection across three diagnostic regimes: (i) low-PRM / correct-final-answer, (ii) low-PRM / incorrect-final-answer, and (iii) high-PRM / correct-final-answer. This design deliberately over-represents cases where PRM scores and final outcomes *disagree*, providing the strongest test of whether the PRM captures step-level validity beyond outcome prediction. Domain experts then blindly annotated every step within the selected trajectories, yielding 194 human-labeled reasoning steps (each labeled as CORRECT, UNCLEAR, or INCORRECT).

As shown in Table 1, PRM scores align strongly with expert judgments: the median score is 0.95 for correct steps versus 0.11 for incorrect steps (0.15 for unclear), achieving an AUROC of 0.94 for separating correct from non-correct steps. Notably, the PRM does not strongly distinguish UNCLEAR from INCORRECT steps; rather, it conservatively assigns both substantially lower scores than correct steps. We also observe that trajectories containing low-PRM intermediate steps but ultimately correct final answers tend to be longer, suggesting the model requires additional steps to recover from earlier unsupported reasoning. Together, these results

*Table 1.* PRM alignment with human expert step-level annotations (194 steps across 50 diagnostic instances). The PRM cleanly separates correct from incorrect reasoning steps.

| Metric | Value |
|---|---|
| Median PRM Score (Correct) | 0.95 |
| Median PRM Score (Unclear) | 0.15 |
| Median PRM Score (Incorrect) | 0.11 |
| AUROC (Correct vs. Non-Correct) | 0.94 |
| Spearman $\rho$ | 0.76 |
| Pairwise Ranking Accuracy | 0.89 |
| ECE | 0.03 |

*Table 2.* Performance comparison of different policy input mode.

| Method | RPE1 (OOD) | | |
|---|---|---|---|
| | DE | DoC | Avg |
| Only Query (Greedy) | 19.65 | 34.84 | 27.25 |
| Only Query (Majority Voting) | 20.69 | 35.86 | 28.28 |
| Only Query (BoN w/ Full Set) | 53.17 | 51.94 | 52.56 |
| SUMMER (Greedy) | 37.52 | 68.85 | 53.19 |
| SUMMER (Majority Voting) | 38.88 | 72.31 | 55.60 |
| SUMMER (BoN w/ Full Set) | 53.06 | 74.95 | 64.01 |

provide direct evidence that the PRM tracks step-level reasoning validity, not merely final-answer correctness.

Together, these evaluations confirm that our PRM is capable of identifying flawed intermediate logic at the individual step level, mitigating the risk of reasoning hallucinations in high-stakes scientific discovery.

### 4.2.3. W2S-INDUCED SAMPLING ENHANCES INSTANCE EFFICIENCY

Figure 3(top) validates the instance-level sampling strategy introduced in Section 3.5.1 on the held-out RPE1 cell line (OOD). Experimental results reveal that W2S-induced sampling exhibits exceptional data efficiency: (1) **Surpassing Full-Set Baselines:** Upon reaching a scale of 100k instances, the W2S-induced model achieves an F1 score of approximately 64.5%, effectively surpassing both the Full Set w/ Multi-label baseline (64.0%) and the Full Set w/ Single-label baseline (62.0%), despite the latter utilizing 351k training instances. (2) **Defining the Pareto-Optimal Frontier:** Across the entire range of data scales, the W2S subsets (indicated by purple markers) consistently maintain F1 scores between 61.3% and 64.6%, firmly establishing the Pareto-optimal frontier. In stark contrast, baseline models such as Llama-PRM800K and Qwen-PRM800K yield F1 scores of only around 57.5%. Such gains suggest that the W2S successfully identifies information-dense training examples, allowing PRM to achieve superior biological reasoning while mitigating the dependence on massive datasets.

### 4.2.4. LABEL-PATTERN-BASED MASKING IMPROVES LABEL EFFICIENCY

We next investigate how pattern-specific masking affects PRM performance and label efficiency (Figure 3(bottom); full results in Appendix D). Our analysis indicates that performance gains are not linearly proportional to the number of supervised steps; instead, selectively retaining high-reliability patterns yields disproportionate gains. For instance, maintaining only the P1 patterns (high SC & high NC) unmasked achieves an F1 score near $\approx 64\%$ using only $\approx 62\%$ of step labels, with a negligible drop relative to the full-label baseline trained with all weak step labels. By contrast, isolating low-reliability regimes (e.g., P3 or P4 alone) produces substantially weaker performance. We also find that *anchoring* neighborhood-reliable steps to high-confidence anchors can further improve generalization: unmasking P3 in addition to P1 (P1+P3) yields higher OOD performance than P1 alone, suggesting that steps which are ambiguous in isolation but reliable within their neighborhood provide complementary supervision when grounded by high-confidence anchors. Overall, these results validate the P1–P4 stratification and show that effective PRM training does not require uniform supervision of every reasoning step.

### 4.3. Generalization Across Prompting Modes, Policy Models and Tasks

To test generalization and robustness beyond our default setting, we evaluate our PRM across input modalities, policy models, and downstream tasks. As shown in Tables 2 and 7, our PRM consistently outperforms greedy decoding and majority voting under both *Only Query* (question only) and *SUMMER* prompting mode (query augmented with retrieved summaries) (Wu et al., 2025), in both IID and OOD settings. Additionally, the consistent gains observed on the rbio1 policy model (Istrate et al., 2025) across cell lines (Tables 3 and 8) further demonstrate that DC-W2S transfers across policy distributions.

Notably, on the BioReason KEGG task, a domain that lies entirely outside the training distribution, the W2S-enhanced models trained with only 100k samples surpass those trained on the full-set counterpart. As reported in Table 4, BoN with CellProfiler (100k) and ESM (100k) achieve Weighted F1 scores of 88.89% and 89.08%, respectively, outperforming the Full Set variant (87.40%). This "less-is-more" effect suggests that training on the full pool of weak labels may introduce considerable noise or conflicting supervision that undermines cross-task transfer. In contrast, our W2S mechanism selectively filters low-quality signals and distills more transferable biological structure, leading to superior generalization on unseen tasks.

*Table 3.* Results of **DC-W2S** using RBIO1 policy model.

| Method | RPE1 (OOD) | | |
|---|---|---|---|
| | DE | DoC | Avg |
| Greedy Decoding | 78.33 | 34.89 | 56.61 |
| Majority Voting | 78.51 | 30.69 | 54.60 |
| Coverage (Upper Bound) | 79.18 | 91.99 | 85.59 |
| **Our PRM** | | | |
| BoN w/ Full Set (Multi-Label) | 78.88 | 62.30 | 70.59 |
| BoN w/ CellProfiler (100k) | 78.82 | 58.04 | 68.43 |
| BoN w/ ESM (100k) | 78.81 | 60.65 | 69.73 |

*Table 4.* Results of **DC-W2S** on BioReason KEGG Task.

| Method | Acc | Weighted F1 |
|---|---|---|
| Greedy Decoding | 80.69 | 87.41 |
| Majority Voting | 82.76 | 87.23 |
| Coverage (Upper Bound) | 95.86 | 97.43 |
| **Our PRM** | | |
| BoN w/ Full Set (Multi-Label) | 83.45 | 87.40 |
| BoN w/ CellProfiler (100k) | 84.48 | 88.89 |
| BoN w/ ESM(100k) | 84.14 | 89.08 |

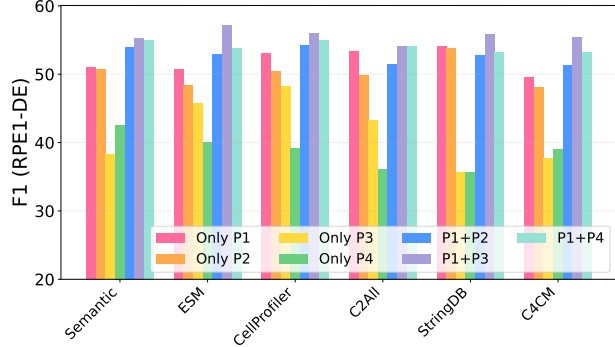

*Figure 4.* **Effect of embedding choice.** Performance on RPE1–DE for different label-pattern configurations across semantic and biologically grounded embedding spaces. Embeddings with richer biological structure (e.g., ESM, CellProfiler) yield larger gains from neighborhood-reliable supervision (P3).

## 5. Discussion

We analyze how weak step supervision and neighborhood geometry jointly shape PRM performance. Overall, the P1–P4 stratification is highly predictive of supervision value. P1 (high SC and high NC) is the most reliable anchor: across models and tasks, `Only P1` is the strongest single-pattern regime, confirming P1 steps as high-fidelity anchors. P3 (low SC but high NC) captures steps that are ambiguous in isolation yet coherent within a biologically meaningful neighborhood; while `Only P3` performs poorly on its own, P3 becomes valuable when anchored to P1, providing complementary supervision that improves generalization.

With P1 as an anchor, the marginal benefit of incorporating other patterns (P2–P4) is setting-dependent. For DoC tasks, which mainly require predicting the *sign* of a gene response, `P1+P2` is typically comparable to or slightly better than `P1+P3`: directional cues align with causal-directional priors encoded in biological text and LLM pretraining, so neighborhood geometry adds limited incremental value. By contrast, for OOD–DE we observe `P1+P3 > P1+P2` consistently across embeddings: DE requires separating meaningful expression changes from statistical and biological noise, so manifold-consistent but teacher-ambiguous cues (P3) provide transferable structure that weak teachers alone do not capture. This is consistent with the expansion-property interpretation of W2S generalization (Lang et al., 2024). Figure 10a shows this "ΔP3 > ΔP2" effect.

Although we expected `P1+P4` to degrade performance, in some settings it provides an average benefit. One plausible explanation is that P4 contains a mixture of truly noisy steps and a minority of informative steps that are *hard* rather than incorrect; adding such steps may act as a mild regularizer

or increase coverage of rare reasoning modes. Nonetheless, P4 should be treated cautiously and is generally a primary candidate for masking or controlled inclusion.

Embedding choice modulates NC. Embedding quality determines whether retrieved neighborhoods reflect biological relatedness (cf. Assumption 3.4). As shown in Figure 4, not all "biological" embeddings are equally helpful (consistent with prior work (Littman et al., 2025)), so neighborhood construction must be validated per task. Embedding effects are most pronounced in biologically demanding or distribution-shifted settings (particularly OOD–DE), where the model cannot rely on simple directional cues or teacher agreement and must instead depend on neighborhood geometry to propagate manifold-consistent structure. Embeddings that encode richer functional or phenotypic signals (e.g., ESM- or CellProfiler-based variants) tend to produce larger gains from P3 than purely semantic or network-only embeddings.

Finally, our theoretical analysis assumes neighborhood label consistency that supports weak-to-strong correction under robust expansion. A natural direction for future work is to study weaker, more local assumptions that directly connect embedding continuity to label correctness (e.g., bounds of the form $\Pr(y(z') \neq y(z_t) \mid z' \in \mathcal{N}(z_t)) \leq \xi$), and to investigate calibrated soft-weighting and conditional reliability models that are pattern- and task-dependent. Another important direction is to empirically validate **DC-W2S** beyond biological perturbation reasoning, by testing whether the same dual-consensus stratification yields similar gains in other complex reasoning domains.

In short, **DC-W2S** shows that consensus signals turn weak supervision into high-value training data, improving PRM performance and label efficiency for biological reasoning.

## Impact Statement

This paper presents work whose goal is to advance the field of Machine Learning, specifically by improving process-level supervision of LLM reasoning in biological settings. If used responsibly, more reliable process reward modeling may improve transparency in scientific workflows and reduce misleading mechanistic rationales. However, the approach inherits limitations from weak supervision and underlying models and may still produce confident but incorrect reasoning traces; it is not a substitute for expert judgment. We encourage domain-specific validation and human-in-the-loop use before any high-stakes deployment.

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

# A. Implementation Details

In this section, we provide exhaustive details regarding our experimental setup, including the data generation process, the technical implementation of our retrieval system, and the training configurations for our PRMs.

## A.1. Weak Supervision and Data Generation

The weak supervision signals are derived from two primary sources: LLM-as-a-judge and Monte Carlo rollouts.

- **LLM-as-a-Judge:** We utilize `Qwen3-32B` as our primary weak supervisor for step-level annotations. The specific prompt templates used for LF-Direct, Context and Analogical are illustrated in Figure 5.

- **Monte Carlo Rollouts:** To obtain ground-truth–oriented weak signals, we perform MC rollouts with the Qwen3 family (1.7B, 4B, and 7B variants). For each intermediate reasoning step, we launch 8 independent rollouts to estimate the probability of ultimately reaching the correct answer.

## A.2. Efficient Neighborhood Retrieval and Biological Refinement

As noted in Section 3.2.1, our training corpus contains 351k reasoning trajectories, which expand to over 4.2M individual reasoning steps. Operating at this scale introduces significant memory demands and retrieval latency during neighborhood-consensus computation. To address these challenges, we adopt a compressed vector indexing strategy outlined below.

**IVF-PQ Indexing Architecture:** To mitigate the "curse of dimensionality" and avoid memory overflow, we leverage an Inverted File Index with Product Quantization (IVF-PQ):

- **IVF Acceleration:** The vector space is partitioned into $K$ Voronoi cells (defined by $num\_clusters = 10,000$). During retrieval, the system probes only a subset of clusters ($nprobe = 32$), reducing the computational complexity of the global search by two orders of magnitude.

- **PQ Compression:** Each 384-dimensional semantic embedding is partitioned into 64 sub-vectors ($pq\_subquantizers = 64$), each quantized into an 8-bit index. This reduces the memory footprint of a single step representation from 1,536 bytes to 64 bytes. This $24\times$ compression allows the entire 4.2M-step index to reside in-memory, facilitating low-latency neighborhood search without frequent disk I/O.

Afterwards, biology-refined neighborhood retrieval is implemented as follows:

1. **Semantic Neighborhood Extraction:** For a query step, we first retrieve $K = 20$ candidates using the IVF-PQ index based on the *all-MiniLM-L6-v2* embedding space.

2. **Biological Manifold Refinement:** For each candidate, we extract the associated gene pair from its query (perturbation gene and target gene). We concatenate their embeddings derived from a biological foundation model. A neighbor is only considered valid for consensus calculation if its biological cosine similarity to the query exceeds a threshold of 0.5. This ensures that the reasoning patterns are not only semantically similar but also grounded in consistent biological contexts.

## A.3. PRM Training Configuration

Our Process Reward Model is initialized from `Qwen3-4B`. We append a regression head to the final transformer layer to predict step-wise rewards. The regression head consists of a multi-layer perceptron (MLP) with a hidden dimension equal to the backbone's hidden size, followed by a ReLU activation and a final linear layer mapping to the reward logits.

**Optimization Hyperparameters:** All models are trained using the AdamW optimizer with a cosine learning rate scheduler and the optimization hyperparameters are as follows.

- **Learning Rate:** 5e-6

- **Global Batch Size:** 256

- **Training Epochs:** 3

- **Weight Decay:** 0.01

- **Warmup Ratio:** 0.1

- **Cutoff Length:** 4096 tokens

**Hardware:** All training experiments are conducted on a NVIDIA B200 cluster.

---

**Algorithm 1** Distribution-Balanced Greedy Subset Selection

---

**Require:** Buckets $\mathcal{B}[p]$ mapping pattern $p$ to sorted list $(i, d_p(i))$ by density $d_p$ descending; Target proportion $\Pi_p = 0.25$;
   Target size $T$

**Ensure:** Selected indices $\mathcal{L}$ and pattern counts $\mathcal{C}$

1: $\mathcal{L} \leftarrow []$
2: $\mathcal{C}[p] \leftarrow 0$ for $p \in \{P1, P2, P3, P4\}$;
3: **for** $t = 1$ to $T$ **do**
4:   Compute deficit $\Delta[p] = \Pi_p - \frac{\mathcal{C}[p]}{\sum_q \mathcal{C}[q] + \epsilon}$
5:   $p^* \leftarrow \arg\max_p \Delta[p]$
6:   $i^* \leftarrow$ first unselected index in $\mathcal{B}[p^*]$
7:   **if** $i^*$ is None **then**
8:     Search all buckets in descending $\Delta[p]$ order for first unselected index; assign to $i^*$
9:   **end if**
10:   **if** $i^*$ is None **then**
11:     **break**
12:   **end if**
13:   Add $i^*$ to $\mathcal{L}$
14:   Update $\mathcal{C}[p]$ for all patterns present in entry $i^*$
15: **end for**
16: **return** $\mathcal{L}, \mathcal{C}$

---

# B. Dataset Details

PerturbQA (Wu et al., 2025) is a structured biological benchmark introduced to assess LLMs' ability to predict discrete outcomes of high-content genetic perturbation experiments and interpret patterns in molecular biology through language. The benchmark is defined over real perturbation data derived from five high-quality single-cell CRISPR interference (CRISPRi) experiments, with ground-truth labels generated through rigorous statistical criteria on differential expression outcomes. PerturbQA defines two binary prediction tasks over perturbation–target gene pairs and cell contexts (e.g., K562, RPE1, HepG2, Jurkat): **DE** (differential expression prediction), which predicts whether a given perturbation induces a significant change in expression for a downstream target gene; and **DoC** (direction of change), a subsequent task that predicts whether the change is an increase or decrease, and is defined only for pairs labeled positive under DE.

BioReason (Fallahpour et al., 2025) is a multimodal biological reasoning benchmark suite. It is specifically designed to evaluate a model's ability to perform multi-step reasoning over genomic sequence data integrated with natural language queries, and its evaluation spans three distinct datasets representing increasingly complex reasoning and classification tasks. The first component, the KEGG-Derived Biological Reasoning Dataset, which we adopt in this work, contains approximately 1,449 examples that connect paired reference/variant DNA sequences to disease phenotypes via mechanistic pathways drawn from curated pathway resources such as KEGG (Kanehisa & Goto, 2000), with reasoning traces that lead from genetic variant context to predicted outcomes. The KEGG dataset is split into train, validation, and test sets with roughly 1,159 training, 144 validation, and 146 test examples, and the input length varies as a function of the genomic and textual context embedded with reasoning annotations.

*Table 5.* Baseline PRM comparison across all cell lines and tasks.

| Method | HepG2 | | Jurkat | | K562 | | RPE1 | | Avg |
|---|---|---|---|---|---|---|---|---|---|
| | DE | DoC | DE | DoC | DE | DoC | DE | DoC | |
| Greedy Decoding | 40.05 | 71.15 | 41.12 | 59.91 | 38.47 | 83.47 | 37.52 | 68.85 | 55.07 |
| Majority Voting | 43.56 | 74.51 | 44.63 | 61.69 | 41.49 | 85.10 | 38.88 | 72.31 | 57.77 |
| **Baselines** | | | | | | | | | |
| BoN w/ Llama-PRM800K | 47.27 | 71.27 | 48.71 | 58.81 | 44.48 | 82.84 | 45.20 | 69.91 | 58.56 |
| BoN w/ Qwen-PRM800K | 44.78 | 73.53 | 45.65 | 61.77 | 41.79 | 84.87 | 42.03 | 72.58 | 58.38 |
| BoN w/ VersaPRM | 46.86 | 69.70 | 47.84 | 57.61 | 43.64 | 80.36 | 43.73 | 67.57 | 57.16 |
| **Our PRM** | | | | | | | | | |
| BoN w/ Full Set | 61.84 | 79.89 | 62.33 | 69.53 | 58.10 | 88.32 | 53.06 | 74.95 | 68.50 |

### B.1. Licenses

We strictly follow all licenses when using the public assets in this work. The PerturbQA dataset and codebase are publicly available under the CC BY 4.0 license and the Genentech Non-Commercial Software License Version 1.0, respectively. The BioReason KEGG dataset is under Apache License, Version 2.0.

## C. Biological Embedding Details

To construct biologically grounded neighborhoods for TNC, we follow (Littman et al., 2025) and use embeddings derived from curated biological knowledge graphs (KGs), pretrained foundation models, and experimental data.

**CellProfiler.** These are gene embeddings derived from optical pooled screening (OPS) experiments, which couple CRISPR perturbations with high-content cellular imaging (Funk et al., 2022). Per-gene morphological profiles are computed from Cell Painting/CellProfiler features, aggregated across perturbations, and PCA-compressed to form gene-level embeddings; we use the precomputed representations released in (Littman et al., 2025).

**ESM.** These are protein-sequence embeddings derived from a pretrained ESM model (Lin et al., 2023); we use the precomputed gene representations released in (Littman et al., 2025).

**C2all and C4CM.** These are network-based embeddings derived from MSigDB database (Liberzon et al., 2011). **C2all** uses the MSigDB C2 curated collection (Canonical Pathways and Chemical/Genetic Perturbations), while **C4CM** uses the MSigDB C4 cancer module collection (Segal et al., 2004). For each collection, gene embeddings are computed by applying node2vec (Grover & Leskovec, 2016) to the corresponding KG.

**StringDB.** These are network-based gene embeddings derived from the STRING database (Szklarczyk et al., 2023), which integrates physical interactions and functional associations. Gene embeddings are computed by applying node2vec (Grover & Leskovec, 2016) to the STRING KG, capturing network proximity and shared interaction neighborhoods.

## D. More Experimental Results

### D.1. Full Results on Generalization and Baselines Comparison

In this section, we provide a comprehensive evaluation of the proposed **DC-W2S** framework, expanding upon the main text results. We benchmark our method against established PRM baselines, analyze the impact of annotator model size, and rigorously test generalization across different policy models (RBIO1), input modes (*Only Query/SUMMER*), and downstream tasks (BioReason KEGG).

*Table 6.* Performance comparison between PRMs trained on different annotation models and using the annotation model (Qwen3-32B) directly as a judge.

| Best-of-N (N=8) | HepG2 | | Jurkat | | K562 | | RPE1 | | |
|---|---|---|---|---|---|---|---|---|---|
| | DE | DoC | DE | DoC | DE | DoC | DE | DoC | Avg |
| Ours w/ Full Set (Annotated by Qwen3-32B) | 61.84 | 79.89 | 62.33 | 69.53 | 58.10 | 88.32 | 53.06 | 74.95 | 68.50 |
| Ours w/ Full Set (Annotated by Qwen3-4B) | 62.04 | 80.11 | 62.86 | 70.07 | 57.41 | 89.32 | 51.42 | 75.86 | 68.63 |
| LLM-as-a-Judge (LF-Analogical, Qwen3-32B) | 36.43 | 64.76 | 38.55 | 53.77 | 33.30 | 82.54 | 32.37 | 64.07 | 50.72 |

*Table 7.* Performance Comparison of Policy Input Modes (IID vs OOD).

| Method | HepG2 | | | Jurkat | | | K562 | | | RPE1 | | | Avg |
|---|---|---|---|---|---|---|---|---|---|---|---|---|---|
| | DE | DoC | Avg | DE | DoC | Avg | DE | DoC | Avg | DE | DoC | Avg | |
| Only Query (Greedy) | 20.26 | 41.37 | 30.82 | 22.03 | 36.37 | 29.20 | 19.23 | 47.37 | 33.30 | 19.65 | 34.84 | 27.25 | 30.14 |
| Only Query (Majority Voting) | 22.40 | 43.45 | 32.93 | 25.98 | 37.01 | 31.50 | 23.08 | 45.68 | 34.38 | 20.69 | 35.86 | 28.28 | 31.77 |
| Only Query (BoN w/ Full Set) | 58.09 | 66.18 | 62.14 | 60.80 | 55.91 | 58.36 | 58.39 | 72.51 | 65.45 | 53.17 | 51.94 | 52.56 | 59.62 |
| SUMMER (Greedy) | 40.05 | 71.15 | 55.60 | 41.12 | 59.91 | 50.52 | 38.47 | 83.47 | 60.97 | 37.52 | 68.85 | 53.19 | 55.07 |
| SUMMER (Majority Voting) | 43.56 | 74.51 | 59.04 | 44.63 | 61.69 | 53.16 | 41.49 | 85.10 | 63.30 | 38.88 | 72.31 | 55.60 | 57.77 |
| SUMMER (BoN w/ Full Set) | 61.84 | 79.89 | 70.87 | 62.33 | 69.53 | 65.93 | 58.10 | 88.32 | 73.21 | 53.06 | 74.95 | 64.01 | 68.50 |

**Comparison against PRM and Strong Annotator Baselines.** As detailed in Table 5, our *BoN w/ Full Set* consistently outperforms all baselines. Notably, on the OOD RPE1 split, our method achieves an average score of 68.50%, significantly surpassing the strongest baseline (*Llama-PRM800K*) which achieves 58.56%. This demonstrates that **DC-W2S** learns a reward function that is not merely memorizing cell-line specific patterns, but capturing the underlying biological reasoning structure, thereby enabling robust transfer to unseen biological contexts. Additionally, a core premise of W2S Generalization is that the student model has the potential to compare or even outperform its supervisor. Table 6 validates this hypothesis. The PRM trained via our framework (Student) achieves an average F1 of 68.50% (using Qwen3-32B annotations) and 68.63% (using Qwen3-4B annotations), both of which substantially outperform the teacher model making prediction itself (LLM-as-a-Judge (LF-Analogical), Qwen3-32B) which scores only 50.72%. Furthermore, the marginal performance difference between using a 32B annotator versus a 4B annotator suggests that our aggregation mechanisms are robust to the quality of the individual weak supervisors, making the framework scalable even with smaller, more efficient annotators.

**Robustness Across Input Modes.** To ensure our gains are not an artifact of specific prompting strategies, we evaluate performance under two distinct input modes: *Only Query* and *SUMMER*. Table 7 shows that while the *SUMMER* setting generally yields higher absolute performance due to retrieved context, our PRM (BoN) consistently provides significant gains over Greedy decoding and Majority Voting in both settings. For instance, in the *Only Query* mode on RPE1, our PRM improves the average score from 27.25% (Greedy) to 52.56%, confirming that the reward model captures intrinsic reasoning validity independent of input context augmentation.

**Transferability to Different Policy Models.** We further assess whether the PRM trained on Qwen3-4B-generated trajectories can generalize to score trajectories from a different policy model, specifically RBIO1 (Istrate et al., 2025). As shown in Table 8, our PRM effectively guides the RBIO1 policy, improving the average RPE1 performance from 55.40% (Greedy) to 73.66% (BoN w/ Full Set). Crucially, the data-efficient variants (*BoN w/ CellProfiler 100k* and *BoN w/ ESM 100k*) achieve performance comparable to the full-set model (e.g., 72.40% vs 73.66% Avg), reinforcing our claim that a curated subset of high-consensus data is sufficient to learn a transferable reward function.

**Cross-Task Generalization on BioReason KEGG.** Additionally, we evaluate **DC-W2S** on a distinct downstream task: the BioReason KEGG pathway prediction, which lies entirely outside the perturbation-based training distribution. Table 9 reveals a compelling finding: while the *BoN w/ Full Set* model yields only marginal gains over greedy decoding, the data-efficient variants achieve even higher performance. This stands in distinct contrast to the policy transfer setting (Table 8), where the data-efficient variants merely achieved parity with the full-set model. This divergence indicates that while the full dataset's noise is tolerable when the downstream task distribution remains close to the source (as with RBIO1), it becomes actively detrimental in cross-task settings. By training indiscriminately on the full dataset, the model overfits to perturbation-specific artifacts; conversely, the **DC-W2S** curated subsets distill a more fundamental biological reasoning signal that is free from task-specific noise, thereby enabling superior generalization to the target KEGG task.

*Table 8.* Detailed Results of **DC-W2S** using RBIO1 policy model.

| Method | HepG2 | | | Jurkat | | | K562 | | | RPE1 | | | Avg |
|---|---|---|---|---|---|---|---|---|---|---|---|---|---|
| | DE | DoC | Avg | DE | DoC | Avg | DE | DoC | Avg | DE | DoC | Avg | |
| Greedy Decoding | 74.72 | 36.04 | 55.38 | 76.01 | 30.56 | 53.28 | 69.79 | 42.88 | 56.34 | 78.33 | 34.89 | 56.61 | 55.40 |
| Majority Voting | 74.97 | 39.20 | 57.08 | 76.10 | 28.74 | 52.42 | 69.96 | 34.47 | 52.21 | 78.51 | 30.69 | 54.60 | 54.08 |
| Coverage (Upper Bound) | 76.18 | 89.92 | 83.05 | 77.02 | 90.94 | 83.98 | 70.87 | 91.12 | 80.99 | 79.18 | 91.99 | 85.59 | 83.40 |
| **Our PRM** | | | | | | | | | | | | | |
| BoN w/ Full Set | 75.65 | 75.09 | 75.37 | 76.43 | 68.70 | 72.56 | 70.46 | 81.79 | 76.13 | 78.88 | 62.30 | 70.59 | 73.66 |
| BoN w/ CellProfiler (100k) | 75.44 | 73.48 | 74.46 | 76.42 | 64.85 | 70.63 | 70.46 | 81.26 | 75.86 | 78.82 | 58.04 | 68.43 | 72.34 |
| $\Delta$ vs Full Set | -0.21 | -1.61 | -0.91 | -0.01 | -3.85 | -1.93 | +0.00 | -0.53 | -0.27 | -0.06 | -4.26 | -2.16 | -1.32 |
| BoN w/ ESM (100k) | 75.56 | 73.58 | 74.57 | 76.40 | 63.57 | 69.98 | 70.43 | 80.25 | 75.34 | 78.81 | 60.65 | 69.73 | 72.40 |
| $\Delta$ vs Full Set | -0.09 | -1.51 | -0.80 | -0.03 | -5.13 | -2.58 | -0.03 | -1.54 | -0.79 | -0.07 | -1.65 | -0.86 | -1.26 |

*Table 9.* Detailed Results of **DC-W2S** on BioReason KEGG Task.

| Method | Acc | Weighted P | Weighted R | Weighted F1 |
|---|---|---|---|---|
| Greedy Decoding | 80.69 | 98.28 | 80.69 | 87.41 |
| Majority Voting | 82.76 | 94.83 | 82.76 | 87.23 |
| Coverage (Upper Bound) | 95.86 | 99.66 | 95.86 | 97.43 |
| **Our PRM** | | | | |
| BoN w/ Full Set | 83.45 | 94.14 | 83.45 | 87.40 |
| BoN w/ CellProfiler (100k) | 84.48 | 96.55 | 84.48 | 88.89 |
| $\Delta$ vs Full Set | +1.03 | +2.41 | +1.03 | +1.49 |
| BoN w/ ESM(100k) | 84.14 | 96.21 | 84.14 | 89.08 |
| $\Delta$ vs Full Set | +0.69 | +2.07 | +0.69 | +1.68 |

*Table 10.* Label-Masking Results of **DC-W2S** on BioReason KEGG Task.

| Method | Acc | Weighted P | Weighted R | Weighted F1 |
|---|---|---|---|---|
| **BoN w/ CellProfiler (100k)** | | | | |
| Only P1 | 82.41 | 95.52 | 82.41 | 87.83 |
| Only P2 | 84.48 | 97.59 | 84.48 | 89.48 |
| Only P3 | 81.72 | 96.21 | 81.72 | 86.30 |
| Only P4 | 80.34 | 94.83 | 80.34 | 85.83 |
| P1 + P2 | 85.17 | 97.59 | 85.17 | 90.18 |
| P1 + P3 | 82.76 | 95.52 | 82.76 | 87.74 |
| P1 + P4 | 85.52 | 97.24 | 85.52 | 90.12 |
| **BoN w/ ESM (100k)** | | | | |
| Only P1 | 78.97 | 96.90 | 78.97 | 86.08 |
| Only P2 | 83.10 | 97.59 | 83.10 | 88.95 |
| Only P3 | 84.14 | 97.59 | 84.14 | 89.46 |
| Only P4 | 85.17 | 95.86 | 85.17 | 89.46 |
| P1 + P2 | 77.24 | 96.90 | 77.24 | 84.67 |
| P1 + P3 | 84.48 | 96.55 | 84.48 | 89.13 |
| P1 + P4 | 86.21 | 96.55 | 86.21 | 90.24 |

**Larger-Scale Proxy Oracle Evaluation.** While the expert study provides high-fidelity validation, its scale (194 steps) is limited by annotation cost. To complement it, we employed stronger model (GPT-5) augmented with biological tools as a proxy oracle for larger-scale evaluation. From 500 sampled questions, we selected candidate trajectories exhibiting the highest PRM score variance as well. Each step was independently evaluated by the proxy oracle as CORRECT or INCORRECT based on biological soundness. We evaluated 4,917 total steps and computed AUROC specifically on 1,346 bio-filtered steps (retaining only steps containing clear biological claims). The step-level AUROC for HepG2, Jurkat, K562, RPE1 is 0.84, 0.82, 0.79, 0.89, respectively. As a result, the PRM maintains strong step-level discriminative ability across all cell lines.

## D.2. Interpretation of Cross-Task Generalization on BioReason KEGG

Together with Table 9 and Table 10, these results show that our framework produces PRMs that meaningfully improve decoding quality under substantial domain shift. We informally interpret the cross-task generalization as follow:

**Kernel Interpretation: Masking Reduces Curvature and Improves Transfer.** A PRM effectively learns a scoring function $f(z_t)$ over reasoning steps in an induced feature space. Training on all P1–P4 steps forces $f$ to interpolate inconsistent labels, resulting in a high-curvature (large RKHS-norm) solution. Such functions generalize poorly under domain shift. DC-W2S behaves like a label-denoised kernel estimator: it trades sample size for smoothness, which is beneficial under distribution shift. Masking and subsampling remove conflicting constraints, yielding a smoother, lower-norm function that is provably more robust out-of-domain. This kernel-based view explains why smaller, filtered PRMs outperform the full-set model on the KEGG task.

### D.3. Full Results on Scaling Trend, Score Aggregation and Metrics

This section details the impact of inference-time compute scaling, the sensitivity of performance to process-reward aggregation strategies, and a comprehensive evaluation across a diverse suite of classification metrics.

**Inference-Time Scaling Trends.** In the main text, we reported Best-of-$N$ (BoN) results at a fixed budget of $N = 8$. Here, we analyze the scaling laws of our PRM by varying $N$ from 2 to 8 across all four cell lines (HepG2, Jurkat, K562, and the OOD RPE1). As illustrated in Figure 11 to Figure 14, we observe a consistent, monotonic improvement in performance as the sampling budget increases.

- **Monotonicity:** The positive correlation between sample coverage $N$ and downstream F1 scores indicates that the PRM provides a well-calibrated ranking signal; it successfully identifies higher-quality trajectories within the stochastic generations of the policy model.

- **OOD Scaling:** Crucially, this scaling trend holds even for the out-of-distribution RPE1 cell line (Figure 14). The *BoN w/ CellProfiler* models continue to extract gains from increased compute, suggesting that the learned reward function generalizes its understanding of "reasoning quality" to unseen biological contexts, rather than merely memorizing training-set answers.

**Impact of Reward Aggregation Strategies.** A PRM produces a sequence of step-wise scores $r(z_t)$, which must be aggregated into a single trajectory score $R(z)$ for ranking. We evaluate three common aggregation functions:

1. **Last:** Using only the score of the final step, $R(z) = r(z_T)$.

2. **Step Lowest:** Using the minimum score across the trajectory, $R(z) = \min_t r(z_t)$. This enforces a "logical bottleneck" assumption, where a chain is only as strong as its weakest step.

3. **Average:** Using the mean of all step scores, $R(z) = \frac{1}{T} \sum_t r(z_t)$, which provides a smoothed estimate of trajectory quality.

4. **Ensemble Majority Voting:** A meta-aggregation strategy that combines the decisions of the three aforementioned functions.

Figure 15 presents the ablation of these strategies on the OOD RPE1 task. We observe that while *Step Lowest* provides a rigorous filter for logical validity, the *Average* and *Last* aggregation generally yields the most robust ranking performance for biological reasoning, balancing the penalty for incorrect steps with the overall coherence of the chain. Additionally, we observe that the *Ensemble Majority Voting* strategy has the potential to provide the superior performance, as it effectively mitigates the inductive biases of individual methods by extracting their consensus.

**Comprehensive Metric Evaluation.** While F1 score is our primary metric given the class imbalance in perturbation tasks (where valid effects are sparse), relying on a single metric can obscure trade-offs between precision and recall. To provide a holistic view of model performance, we present radar charts in Figure 16 - Figure 19 covering seven distinct metrics: Accuracy, Balanced Accuracy, Matthews Correlation Coefficient (MCC), Recall, True Negative Rate (TNR), Precision, and AUC-ROC.

- **Holistic Improvement:** The radar charts demonstrate that DC-W2S (represented by the *Full Set w/ Multi-label* contours) expands the performance envelope across all axes compared to Greedy Decoding and Majority Voting.

- **Robustness to Imbalance:** Notably, we observe significant gains in *MCC* and *Balanced Accuracy*. Since these metrics are resilient to class imbalance, their improvement confirms that our PRM is not merely exploiting prior class probabilities (e.g., predicting "no effect" frequently) but is genuinely improving the separability between successful and failed reasoning trajectories.

### D.4. More Discussion on Label Pattern Effect

We further analyze how weak step supervision and neighborhood geometry shape PRM performance by training PRMs under different label-pattern configurations (P1–P4) and comparing results across seven embedding spaces (Figure 6 - Figure 9).

**P1 provides stable anchor supervision.**   Across models and tasks, `Only P1` is consistently the strongest single-pattern regime (Figure 6 - Figure 9 and Figure 10b), supporting the view that steps with both high SC and high NC act as high-fidelity anchors; they form the foundation on which DC-W2S builds.

**P2 provides strong but locally brittle supervision.**   P2 (high SC, low NC) is often useful in-distribution but less stable under shift: training on `Only P2` achieves 65.68% (IID) and 60.56% (OOD), a 5.12-point drop. When anchored with P1, P2 yields additional gains (OOD: 64.155% vs. 63.40% for `Only P1`) and reduces the IID→OOD drop (4.54 points), suggesting that P1 anchors stabilize supervision from locally brittle regions while expanding coverage beyond the most reliable regime.

**P3 provides neighborhood-consistent supervision that is more shift-robust.**   P3 (low SC, high NC) is weak in absolute terms when used alone, but exhibits notably smaller degradation under distribution shift: training on `Only P3` achieves 59.37% (IID) and 57.52% (OOD), only a 1.86-point drop. This pattern is consistent with the interpretation that P3 captures manifold-consistent cues that transfer across contexts, even when weak teachers disagree pointwise. Importantly, when combined with reliable anchors, P3 yields the strongest OOD gains among non-P1 regimes: `P1+P3` reaches 64.66% on OOD (vs. 64.16% for `P1+P2` and 63.40% for `Only P1`), while also slightly reducing the IID→OOD drop (4.45 points for `P1+P3` vs. 4.54 for `P1+P2`). Across biologically challenging settings (e.g., RPE1–DE), we consistently observe `P1+P3` > `P1+P2`, suggesting that neighborhood reliability supplies complementary structure beyond teacher agreement and is especially valuable under shift (Figure 10a).

**P4 provides low-value supervision and is largely non-transferable.**   P4 (low SC, low NC) performs poorly when used in isolation (`Only P4`: 55.04% IID, 52.40% OOD), consistent with the interpretation that these steps lack both pointwise agreement and neighborhood support. When combined with P1 anchors, P4 is largely neutral: `P1+P4` achieves 64.23% OOD, comparable to `P1+P2` (64.16%) and slightly above `Only P1` (63.40%). Accordingly, for label efficiency, P4 is a natural first choice to mask, as it provides limited standalone signal while having little effect once anchored by P1.

**Empirical validation of consensus-based stratification.**   To directly validate whether our dual-consensus stratification identifies genuinely high-quality supervision, we computed the correlation between human expert annotations and the weak labels assigned to steps in each reliability pattern. P1 steps exhibit near-perfect agreement with expert judgments (0.97), while P4 steps show only weak correlation (0.30). The results for P2 and P3 are 0.87 and 0.39 respectively. This confirms that our unsupervised stratification effectively isolates high-fidelity training signals without requiring ground-truth step labels.

**In directional reasoning tasks (DoC), P1+P2 and P1+P3 perform similarly.**   For both IID and OOD DoC tasks, `P1+P2` is typically comparable to or slightly better than `P1+P3`. Unlike DE, DoC only requires predicting the *sign* of a gene's response, which aligns closely with causal-directional priors encoded in biological text and LLM pretraining (e.g., "activates", "suppresses", "inhibits"). Because directional reasoning relies less on subtle manifold geometry, neighborhood-consistency (P3) adds limited incremental value over high-consensus patterns such as P2. As a result, P3 is most valuable in fine-grained biological discrimination tasks (e.g., DE) or under distribution shift (e.g., RPE1), whereas P2 is often sufficient for simpler directional tasks like DoC.

**Biologically grounded embeddings improve neighborhood reliability.**   The embedding choice directly affects the quality of NC by determining whether retrieved neighbors are biologically meaningful. Two consistent trends emerge. (i) In purely semantic space, `Only P4` can outperform `Only P3`, suggesting that text-based proximity may retrieve linguistically similar but biologically mismatched steps; in this case, neighborhood reliability estimates become noisy and P3 is less informative. (ii) By contrast, `Only P3` and `P1+P3` generally improve when neighborhoods are built from embeddings with stronger biological structure (e.g., ESM, CellProfiler). Overall, these results indicate that the DC-W2S mechanism is universal, but its gains are amplified when the embedding space provides a biologically coherent geometry for identifying reliable step neighbors.

**When does embedding choice matter most?**   As shown in Figure 10, embedding choice matters most in biologically demanding or distribution-shifted settings (especially OOD–DE), where models cannot rely on simple directional cues or teacher agreement and must instead use neighborhood geometry to propagate NC-driven structure. In these regimes, biologically grounded embeddings (e.g., ESM, C4CM, CellProfiler) yield larger gains from incorporating P3 than semantic

or network-only embeddings. By contrast, in DoC or near-IID settings (e.g., DE-IID), P1 dominates and differences across embeddings are smaller.

---

**Prompt For LF-Direct, Context, Analogical**

You are an expert molecular biologist who studies how genes are related using Perturb-seq and other functional
    genomics approaches. Your task is to evaluate the correctness of intermediate reasoning steps in
    biological analyses, particularly those involving gene regulatory networks, causal inference, and
    molecular mechanisms.

You will be presented with step-by-step reasoning chains where a model attempts to solve complex biological
    problems. For each evaluation, you will receive:

1. The initial question and the step-by-step reasoning chain to evaluate
2. The ground truth answer to the problem
3. Relevant information containing established biological relationships, pathways, and molecular interactions.

Your job is to assess each individual step for:

1. Correctness: Is the biological reasoning scientifically accurate? Are the cited mechanisms, pathways, or
    relationships supported by the established biological knowledge?
2. Logical validity: Does the step follow logically from the previous information? Are there any logical gaps
    or non sequiturs?
3. Alignment with ground truth: Does the step move the reasoning toward or away from the correct answer? Note
    that a step can be scientifically correct but still misaligned with the optimal reasoning path.

For each step, provide:

1. A judgment using one of three categories:
CORRECT: The step contains accurate biological reasoning and contributes meaningfully to solving the problem
NEUTRAL: The step serves a structural/transitional role (connecting ideas) without making substantive claims
    that could be right or wrong, but maintains coherence
INCORRECT: The step contains factual errors, logical flaws, or misleading reasoning
2. A confidence score (1-5, where 5 is most confident)
3. A brief explanation (1-2 sentences) justifying your assessment

Pay special attention to common pitfalls such as:

1. Confusing correlation with causation in gene expression data
2. Oversimplifying complex regulatory networks
3. Misinterpreting statistical significance in genomics contexts
4. Making claims about directionality without appropriate causal inference methods
5. Ignoring cell-type specificity or temporal dynamics
6. Contradicting established relationships in the knowledge graph

Remember that transitional sentences that maintain logical flow should typically be labeled as NEUTRAL rather
    than forced into CORRECT/INCORRECT categories.

Here is the relevant information:

{LF-Direct, Context, and Analogical are only different here given the provided context.}

Now, please evaluate the following reasoning chain and make sure your output is a valid JSON object:

Question:
{instruction}

Reasoning Chain:
{reasoning_chain_with_prefix}

Ground truth:
{label_string}

*Figure 5.* The exact prompt template used for weak supervision generation via LLM-as-a-judge. The blue text indicates where the injected context differs across the three methods (LF-Direct, Context, Analogical).

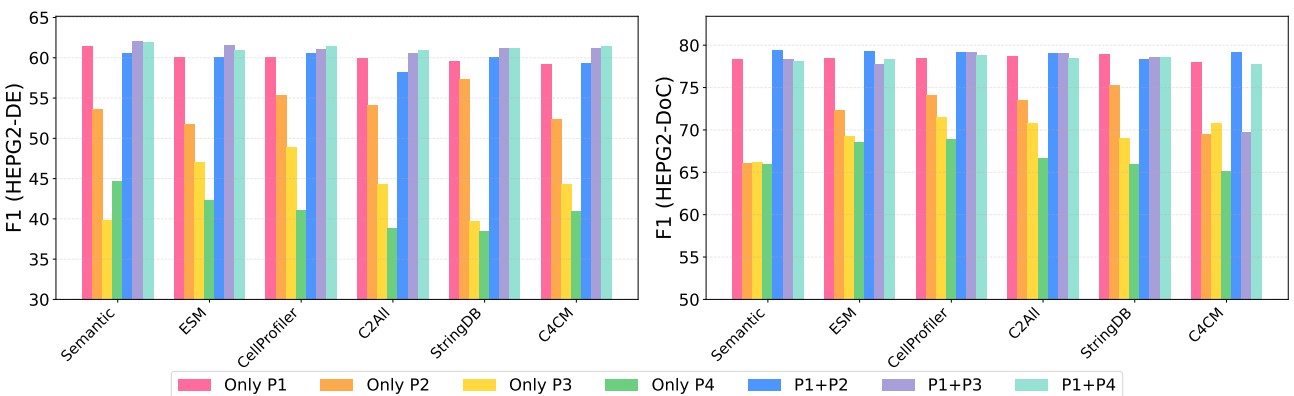

*Figure 6.* Performance Comparison of Different Gene Embedding Approaches (HEPG2).

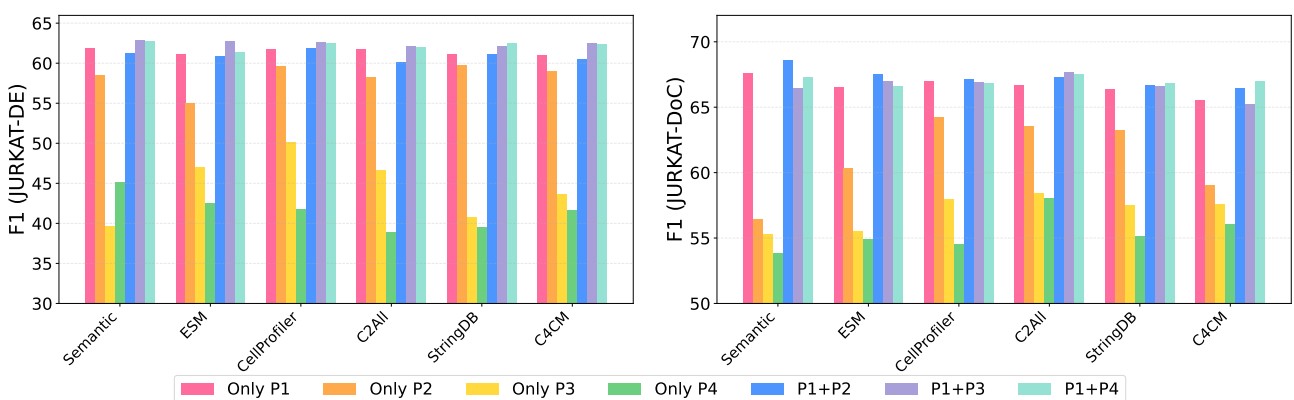

*Figure 7.* Performance Comparison of Different Gene Embedding Approaches (JURKAT).

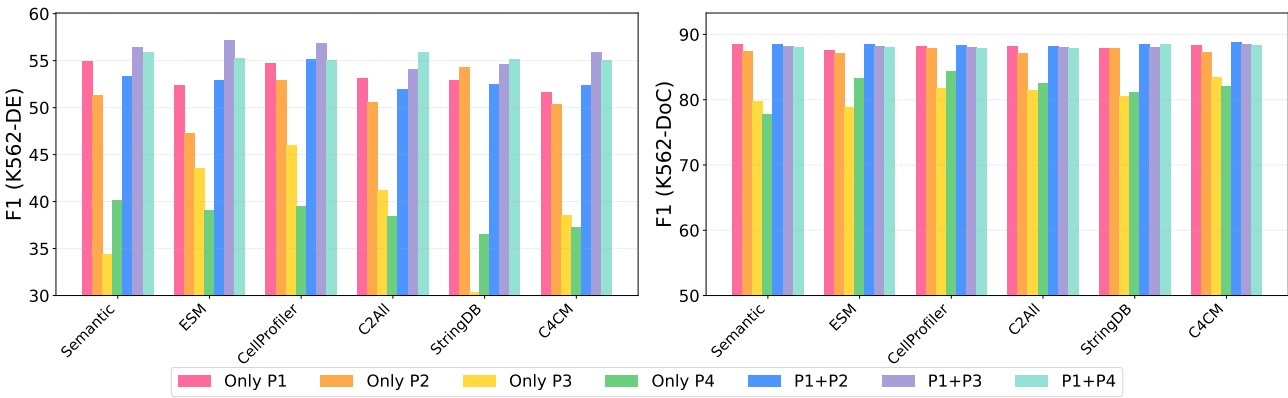

*Figure 8.* Performance Comparison of Different Gene Embedding Approaches (K562).

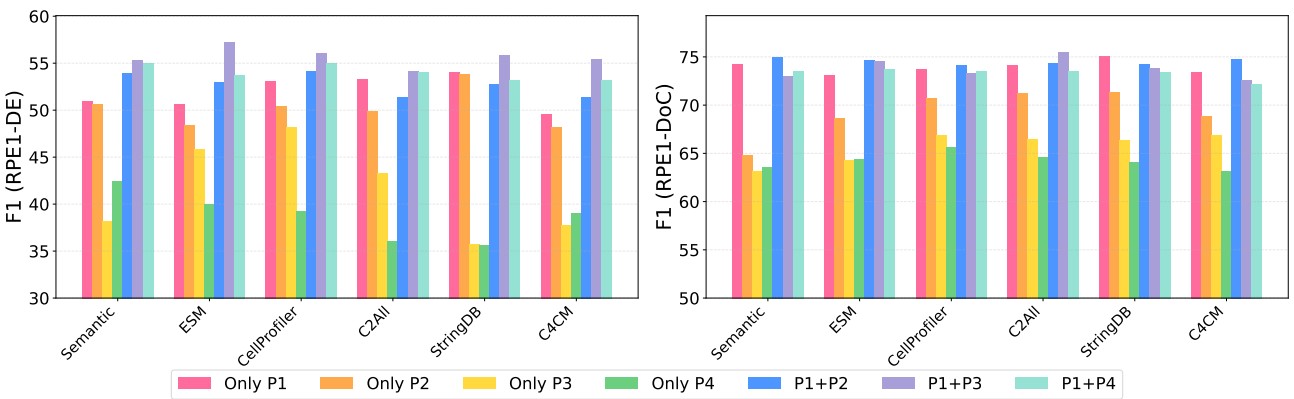

*Figure 9.* Performance Comparison of Different Gene Embedding Approaches (RPE1).

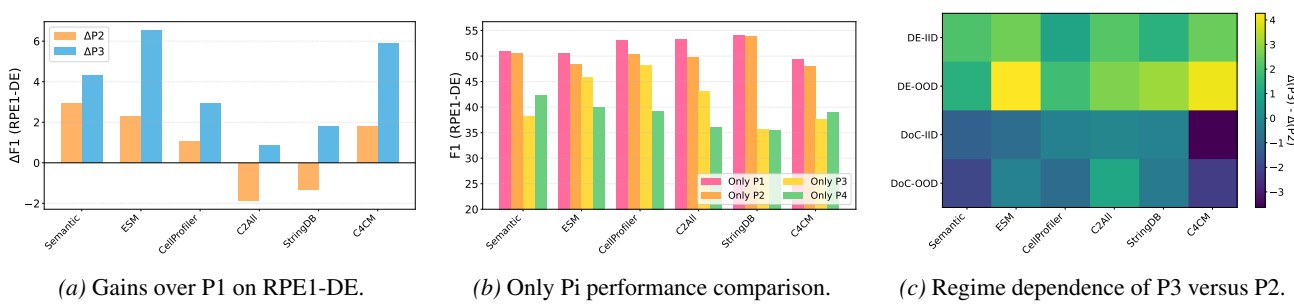

*(a)* Gains over P1 on RPE1-DE.   *(b)* Only Pi performance comparison.   *(c)* Regime dependence of P3 versus P2.

*Figure 10.* **Comparison of label pattern across embeddings and regimes.**

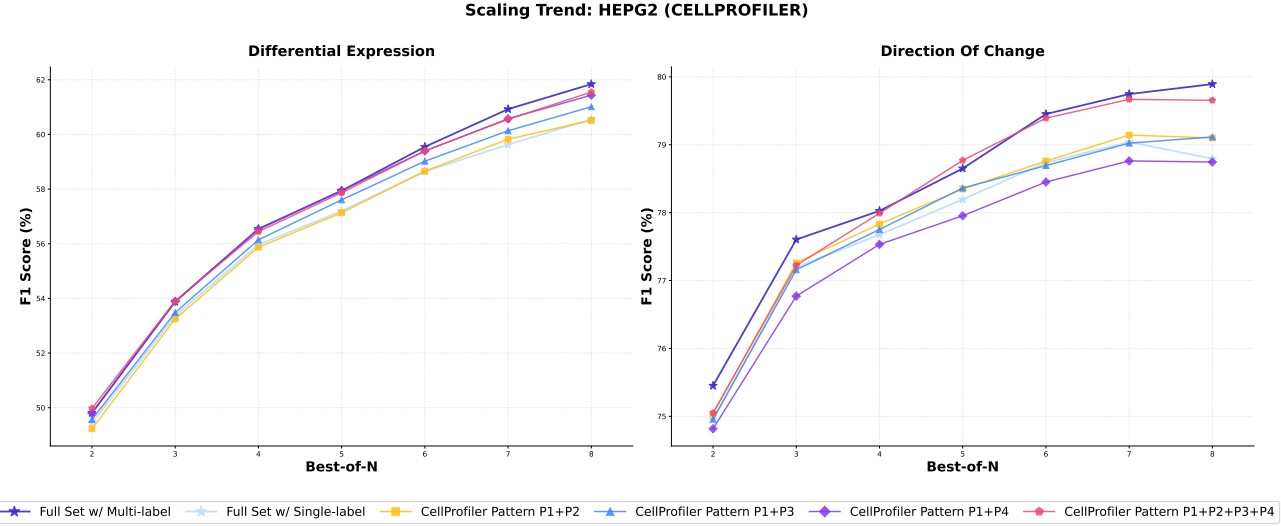

*Figure 11.* Scaling trend of BoN (w/ CellProfiler) on HEPG2.

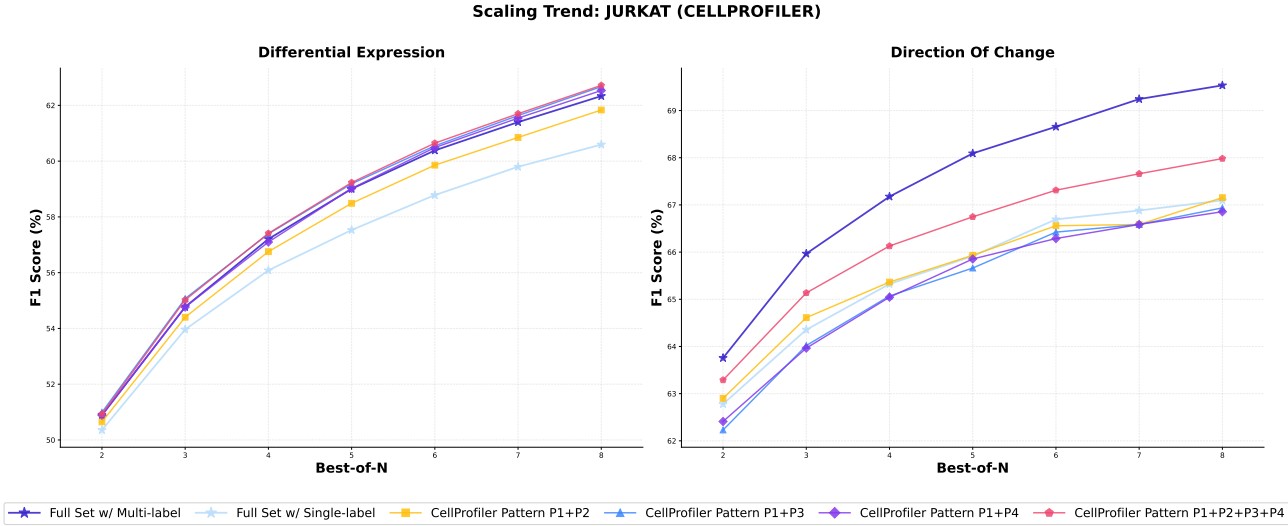

*Figure 12.* Scaling trend of BoN (w/ CellProfiler) on JURKAT.

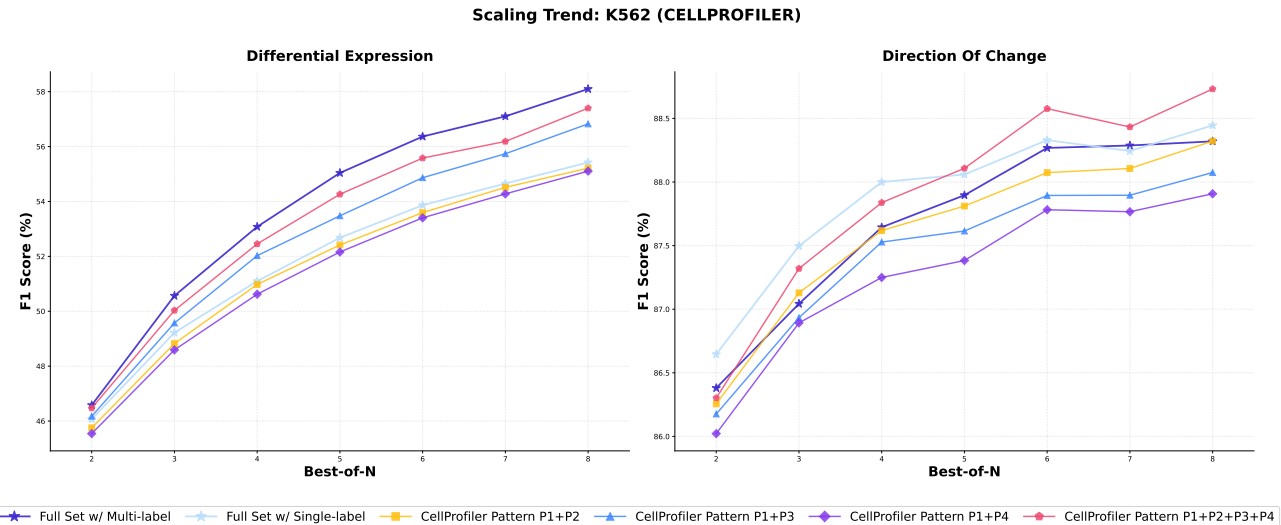

*Figure 13.* Scaling trend of BoN (w/ CellProfiler) on K562.

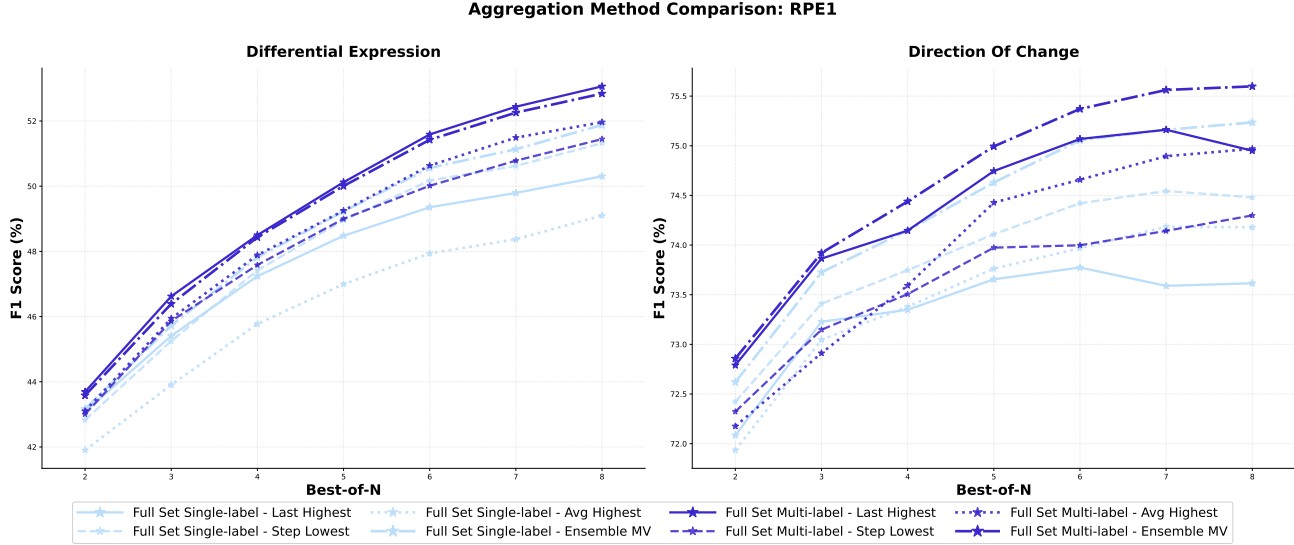

*Figure 14.* Scaling trend of BoN (w/ CellProfiler) on RPE1.

*Figure 15.* Aggregation method comparison on BoN (w/ Full Set) on RPE1(OOD).

**Model Performance: HEPG2**

*Figure 16.* Results of additional metrics on HEPG2.

**Model Performance: JURKAT**

*Figure 17.* Results of additional metrics on JURKAT.

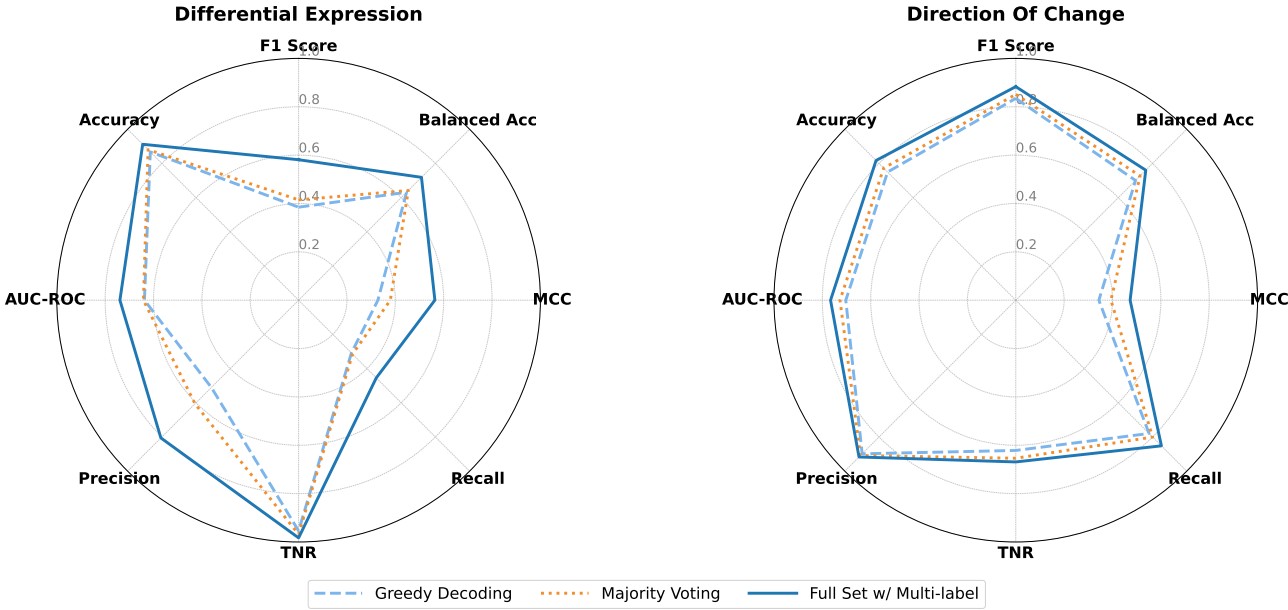

*Figure 18.* Results of additional metrics on K562.

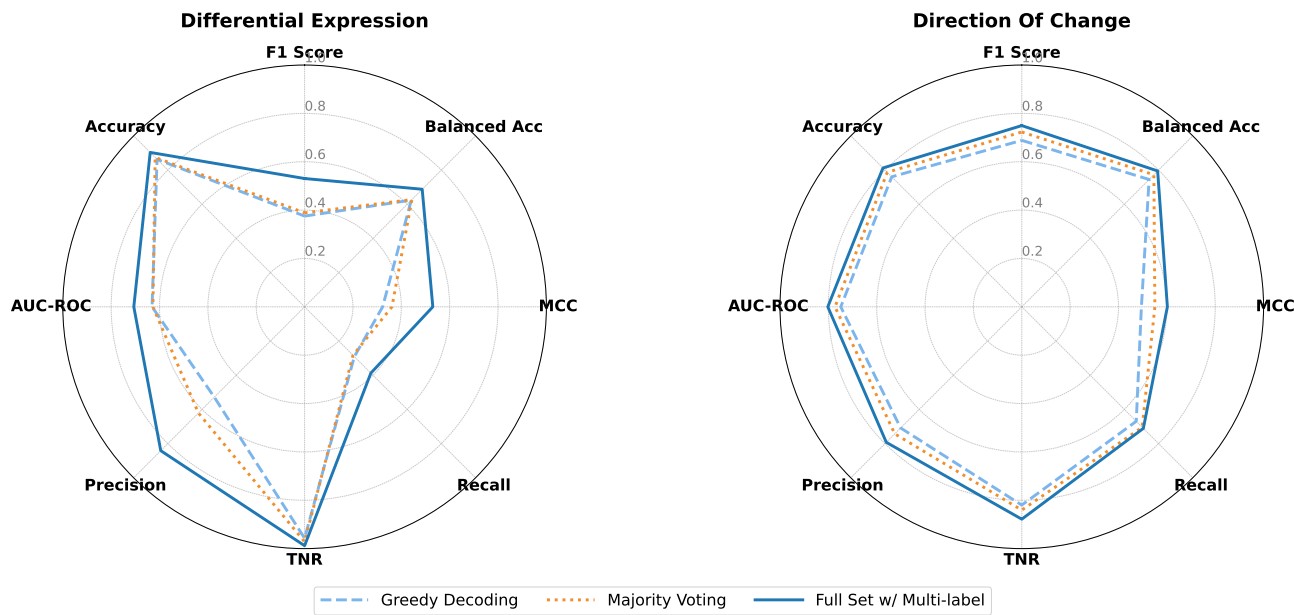

*Figure 19.* Results of additional metrics on RPE1.

# E. Additional theory and proofs

### E.1. Proof of Theorem 3.5

In Section 3.6, we show that under calibrated reliability weights, soft robust expansion, and neighborhood robustness, the *ground truth* step-label error of the thresholded PRM can be controlled by its weak-label error plus terms depending on the effective noise level and the mass of non-robust points.

**Latent correctness indicator.** Since the true step label $y(z_t)$ is unobserved, we assume there exists a latent binary variable such that

$$h(z_t) = \mathbf{1}\{\tilde{y}_{\mathrm{agg}}(z_t) = y(z_t)\} \in \{0,1\}.$$

By Assumption 3.1, $\Pr(h(z_t) = 1 \mid z_t) = p(z_t)$ and $\Pr(h(z_t) = 0 \mid z_t) = 1 - p(z_t)$, hence $\alpha = \Pr(h(z_t) = 0)$. Moreover, for any measurable $U$,

$$\mu_{\mathrm{good}}(U) = \mathbb{E}[p(z_t)\mathbf{1}\{z_t \in U\}] = \Pr(z_t \in U,\, h(z_t) = 1),$$
$$\mu_{\mathrm{bad}}(U) = \mathbb{E}[(1 - p(z_t))\mathbf{1}\{z_t \in U\}] = \Pr(z_t \in U,\, h(z_t) = 0).$$

**Error and correctness sets.** Let the mistake sets of $r_\theta(z_t)$ on the true latent step labels and weak aggregated step labels be $M = \{z_t : f_\tau(z_t) \neq y(z_t)\}$ and $D = \{z_t : f_\tau(z_t) \neq \tilde{y}_{\mathrm{agg}}(z_t)\}$, respectively. Define the set of steps where the PRM decision matches the true step label,

$$G = \mathcal{Z} \setminus M = \{z_t : f_\tau(z_t) = y(z_t)\}.$$

The latent indicator $h(z_t)$ links these two notions: when $h(z_t) = 1$, the weak step label equals the true step label, so any true-label mistake must also be a weak-label disagreement,

$$M \cap \{h = 1\} \subseteq D. \tag{6}$$

Conversely, when $h(z_t) = 0$, the weak label is incorrect, so any true-label correct prediction necessarily disagrees with the weak label,

$$G \cap \{h = 0\} \subseteq D. \tag{7}$$

**A large robust anchor set.** We focus on steps that are (i) locally robust under $\mathcal{N}(\cdot)$, (ii) correctly predicted by $r_\theta$ with respect to the true (latent) step label, and (iii) correctly weak-labeled by aggregation. Define

$$V = R_\eta(f_\tau) \cap G \cap \{h = 1\}.$$

Since $V \subseteq \{h = 1\}$, we have $\mu_{\mathrm{good}}(V) = \Pr(V)$. We claim that $\mu_{\mathrm{good}}(V) \geq q$. Indeed,

$$\Pr(V) = \Pr(h = 1) - \Pr\Big(\big(\{f_\tau \neq y\} \cup \{z_t \notin R_\eta(f_\tau)\}\big) \cap \{h = 1\}\Big)$$
$$\geq (1 - \alpha) - \Pr\Big(\big(\{f_\tau \neq y\} \cup \{z_t \notin R_\eta(f_\tau)\}\big)\Big)$$
$$\geq (1 - \alpha) - \Pr\Big(D \cup \{z_t \notin R_\eta(f_\tau)\}\Big),$$

where the second inequality uses (6), i.e., $(\{f_\tau \neq y\} \cap \{h = 1\}) \subseteq D$. By the theorem premise, $\Pr\big(D \cup z_t \notin R_\eta(f_\tau)\big) \leq 1 - q - \alpha$, so $\Pr(V) \geq q$, and therefore $\mu_{\mathrm{good}}(V) \geq q$. This allows us to invoke the robust expansion assumption.

**Apply soft robust expansion.** Since $V \subseteq R_\eta(f_\tau)$ and $\mu_{\mathrm{good}}(V) \geq q$, $(c, q, \eta)$-soft robust expansion yields

$$\mu_{\mathrm{bad}}(\mathcal{N}_{\mathrm{stab}}(V)) \geq c\,\mu_{\mathrm{good}}(V). \tag{8}$$

**A prediction-stable neighborhood.** To convert (8) into a lower bound on the mass of *corrected* weak labels, we restrict to neighbors that inherit the PRM decision. For any set $U$, define

$$\mathcal{N}_{\mathrm{stab}}(U) := \big\{u' : \exists u \in U,\ u' \in \mathcal{N}(u) \text{ and } f_\tau(u') = f_\tau(u)\big\}.$$

Let

$$B = G \cap \{h = 0\} = \{z_t : f_\tau(z_t) = y(z_t) \text{ and } \tilde{y}_{\mathrm{agg}}(z_t) \neq y(z_t)\}$$

be the set of steps where the weak label is incorrect but the PRM decision is correct.

**Stable neighbors of anchors are ground truth correct.** Take any $u' \in \mathcal{N}_{\mathrm{stab}}(V)$. By definition, there exists $u \in V$ such that $f_\tau(u') = f_\tau(u)$. Since $u \in V \subseteq G$, we have $f_\tau(u) = y(u)$. Moreover, because $\mathcal{N}(\cdot)$ is label-consistent on robust anchors (Assumption 3.4 in the main text), we have $y(u') = y(u)$. Therefore

$$f_\tau(u') = f_\tau(u) = y(u) = y(u'),$$

so $u' \in G$. Hence,

$$\mathcal{N}_{\mathrm{stab}}(V) \subseteq G. \tag{9}$$

Therefore

$$
\begin{aligned}
\mu_{\mathrm{bad}}(G) &\geq c\,\mu_{\mathrm{good}}(V) \\
&= \frac{c}{1-\alpha}\Pr(V \cap \{h=1\}) \\
&= \frac{c}{1-\alpha}\Pr(R_\eta(f_\tau) \cap G \cap \{h=1\}) \\
&= \frac{c}{1-\alpha}\left(\Pr(R_\eta(f_\tau) \cap G) - \Pr(R_\eta(f_\tau) \cap G \cap \{h=0\})\right)
\end{aligned}
$$

Thus

$$
\begin{aligned}
\frac{c}{1-\alpha}\Pr(R_\eta(f_\tau) \cap G) &\leq \mu_{\mathrm{bad}}(G) + \frac{c}{1-\alpha}\Pr(R_\eta(f_\tau) \cap G \cap \{h=0\}) \\
&= \mu_{\mathrm{bad}}(G) + \frac{c\alpha}{1-\alpha}\mu_{\mathrm{bad}}(R_\eta(f_\tau) \cap G) \\
&= \mu_{\mathrm{bad}}(G) + \frac{c\alpha}{1-\alpha}\left(\mu_{\mathrm{bad}}(G) - \mu_{\mathrm{bad}}(G \cap \bar{R}_\eta(f_\tau))\right) \\
&= \mu_{\mathrm{bad}}(G)\left(1 + \frac{c\alpha}{1-\alpha}\right) - \frac{c\alpha}{1-\alpha}\mu_{\mathrm{bad}}(G \cap \bar{R}_\eta(f_\tau)).
\end{aligned}
$$

Therefore

$$\mu_{\mathrm{bad}}(G) \geq \frac{c}{1-\alpha+c\alpha}\Pr(R_\eta(f_\tau) \cap G) + \frac{c\alpha}{1-\alpha+c\alpha}\mu_{\mathrm{bad}}(G \cap \bar{R}_\eta(f_\tau)) \tag{10}$$

**Lemma E.1.** $(M \cap \{h=1\}) \cup (U \cap \{h=0\}) \subset D.$

*Proof.* Let $z_t$ be an arbitrary element of $(M \cap \{h=1\}) \cup (G \cap \{h=0\})$. We consider two cases.

**Case 1:** $z_t \in M \cap \{h=1\}$. Since $z_t \in M$, by definition of $M$ we have $f_\tau(z_t) \neq y(z_t)$. Moreover, since $z_t \in \{h=1\}$, $y_{\mathrm{agg}}(z_t) = y(z_t)$. Therefore, $f_\tau(z_t) \neq y_{\mathrm{agg}}(z_t)$, which implies $z_t \in D$.

**Case 2:** $z_t \in G \cap \{h=0\}$. Since $z_t \in G$, by definition of $G$ we have $f_\tau(z_t) = y(z_t)$. Since $z_t \in \{h=0\}$, $y_{\mathrm{agg}}(z_t) \neq y(z_t)$. Hence, $f_\tau(z_t) \neq y_{\mathrm{agg}}(z_t)$, and thus $z_t \in D$.

In both cases, an arbitrary element of $(M \cap \{h=1\}) \cup (G \cap \{h=0\})$ belongs to $D$. $\qquad\square$

According to the E.1,

$$
\begin{aligned}
\Pr(D) &\geq \Pr(M \cap \{h=1\}) + \Pr(G \cap \{h=0\}) \\
&= (1-\alpha)\mu_{\mathrm{good}}(M) + \alpha\mu_{\mathrm{bad}}(G)
\end{aligned}
$$

Using (10) into the above inequality yields

$$\Pr(D) \geq (1-\alpha)\mu_{\mathrm{good}}(M) + c'\alpha\left(\Pr(R_\eta(f_\tau) \cap G) + \alpha\mu_{\mathrm{bad}}(G \cap \bar{R}_\eta(f_\tau))\right) \tag{11}$$

Using the definition of $G$, we have $\Pr(G) = \alpha\mu_{\mathrm{bad}}(G) + (1-\alpha)\mu_{\mathrm{good}}(G)$. Combining this with (10), we obtain

$$\Pr(G) \geq c'\alpha\left(\Pr(R_\eta(f_\tau) \cap G) + \alpha\mu_{\mathrm{bad}}(G \cap \bar{R}_\eta(f_\tau))\right) + (1-\alpha)\mu_{\mathrm{good}}(G).$$

Therefore,

$$\mu_{\text{good}}(G) \leq \frac{1}{1-\alpha} \left( \Pr(G) - c'\alpha \left( \Pr(R_\eta(f_\tau) \cap G) + \alpha\mu_{\text{bad}}(G \cap \bar{R}_\eta(f_\tau)) \right) \right). \tag{12}$$

Combining this with the definition of set $G$, we have

$$
\begin{aligned}
\mu_{\text{good}}(M) &= 1 - \mu_{\text{good}}(G) \\
&\geq 1 - \frac{1}{1-\alpha} \left( \Pr(G) - c'\alpha \left( \Pr(R_\eta(f_\tau) \cap G) + \alpha\mu_{\text{bad}}(G \cap \bar{R}_\eta(f_\tau)) \right) \right).
\end{aligned} \tag{13}
$$

Using (13) in (11), we obtain

$$
\begin{aligned}
\Pr(D) &\geq (1-\alpha) - \Pr(G) + 2c'\alpha \left( \Pr(R_\eta(f_\tau) \cap G) + \alpha\mu_{\text{bad}}(G \cap \bar{R}_\eta(f_\tau)) \right) \\
&\geq (1-\alpha) - (1 - \Pr(M)) + 2c'\alpha \left( 1 - \Pr(M \cup \bar{R}_\eta(f_\tau)) + \Pr(G \cap \bar{R}_\eta(f_\tau) \cap \{h=0\}) \right) \\
&= \Pr(M) + \alpha(2c'-1) + 2c'\alpha \left( \Pr(G \cap \bar{R}_\eta(f_\tau) \cap \{h=0\}) - \Pr(M \cup \bar{R}_\eta(f_\tau)) \right) \\
&\geq \Pr(M) + \alpha(2c'-1) - 2c'\alpha \Pr(M \cup \bar{R}_\eta(f_\tau)) \\
&= \Pr(M) + \alpha(2c'-1) - 2c'\alpha \left( \Pr(M) + \bar{\rho}_\eta \right) \\
&= (1 - 2c'\alpha) \Pr(M) + \alpha(2c'-1) - 2c'\alpha\bar{\rho}_\eta.
\end{aligned} \tag{14}
$$

**Relate ground truth error to weak error and finish.** Since $\Pr(M) = \text{err}_\tau(r_\theta, y)$ and $\Pr(D) = \text{err}_\tau(r_\theta, \tilde{y}_{\text{agg}})$ are ground truth and weak errors, respectively, we have

$$\text{err}_\tau(r_\theta, y) \leq \frac{\text{err}_\tau(r_\theta, \tilde{y}_{\text{agg}}) - \alpha(2c'-1) + 2c'\alpha\,\bar{\rho}_\eta}{1 - 2c'\alpha}.$$

### E.2. Robust Generalization and Effective Complexity under DC-W2S

So far, our analysis has focused on the bounding step-label error of $r_\theta$. We now complement this with a standard generalization bound that quantifies how neighborhood smoothness controls robust risk via a Rademacher-type complexity.

**Robust risk and empirical robust risk.** Let $\ell : [0,1] \to [0,1]$ be a 1-Lipschitz loss in its first argument (e.g., squared loss or logistic loss applied to $f$ and a weak label $\tilde{y}_{\text{agg}}(z_t)$). Given a neighborhood operator $\mathcal{N}(z_t)$, we define the *robust loss* at step $z_t$ as

$$\tilde{\ell}(f_\tau, z_t) \triangleq \sup_{z' \in \text{supp}(\mathcal{N}(z_t))} \ell\left(f_\tau(z'), \tilde{y}_{\text{agg}}(z')\right), \tag{15}$$

where $\tilde{y}_{\text{agg}}(z')$ is the (possibly noisy) teacher supervision at $z'$. Assume local label consistency inside neighborhoods $\forall z_t, \forall z' \in \text{supp}(\mathcal{N}(z_t)) : \tilde{y}_{\text{agg}}(z') = \tilde{y}_{\text{agg}}(z_t)$, so that $\ell(f_\tau(z'), \tilde{y}_{\text{agg}}(z')) = \ell(f_\tau(z'), \tilde{y}_{\text{agg}}(z_t))$, i.e. we only adversarially perturb the step within its neighborhood but keep the supervision for that base step fixed. The corresponding *robust risk* under the data distribution $\mathcal{D}$ is

$$R^{\text{rob}}(f_\tau) \triangleq \mathbb{E}_{z_t \sim \mathcal{D}}\left[\tilde{\ell}(f_\tau, z_t)\right], \tag{16}$$

and given an i.i.d. sample $\{z_t^{(1)}, \ldots, z_t^{(n)}\}$ from $\mathcal{D}$, the empirical robust risk is

$$\hat{R}_n^{\text{rob}}(f_\tau) \triangleq \frac{1}{n} \sum_{i=1}^{n} \tilde{\ell}(f_\tau, z_t^{(i)}). \tag{17}$$

**Robust Rademacher complexity.** For a function class $\mathcal{F}$ of PRMs $f_\tau : \mathcal{Z} \to [0,1]$, we define the *robust empirical process* via

$$\mathfrak{R}_n^{\text{rob}}(\mathcal{F}) \triangleq \mathbb{E}_{Z,\sigma}\left[\sup_{f_\tau \in \mathcal{F}} \frac{1}{n} \sum_{i=1}^{n} \sigma_i \tilde{\ell}(f_\tau, z_t^{(i)})\right], \tag{18}$$

where $Z = \{z_t^{(1)}, \ldots, z_t^{(n)}\}$ with $z_t^{(i)} \sim \mathcal{D}$ and $\sigma_1, \ldots, \sigma_n$ are i.i.d. Rademacher variables. Analogously, the *standard* (non-robust) Rademacher complexity is

$$\mathfrak{R}_n(\mathcal{F}) \triangleq \mathbb{E}_{Z,\sigma}\left[\sup_{f_\tau \in \mathcal{F}} \frac{1}{n} \sum_{i=1}^{n} \sigma_i \ell(f_\tau, z_t^{(i)})\right]. \tag{19}$$

We now specialize to a class of *pointwise robust PRMs*.

**Definition E.2** (Pointwise $\eta$-robust hypothesis class). For $\eta \geq 0$, let $\mathcal{F}_\eta$ be the set of PRMs $f_\tau : \mathcal{Z} \to [0, 1]$ such that for all $z_t \in \mathcal{Z}$ and all $z' \in \mathrm{supp}(\mathcal{N}(z_t))$,

$$|f_\tau(z') - f_\tau(z_t)| \leq \eta. \tag{20}$$

This pointwise robustness allows us to relate robust and non-robust complexities.

**Lemma E.3** (Robust complexity is controlled by standard complexity). *Assume $\ell$ is 1-Lipschitz in its first argument and bounded in $[0, 1]$. Then for the robust class $\mathcal{F}_\eta$ in Definition E.2,*

$$\mathfrak{R}_n^{\mathrm{rob}}(\mathcal{F}_\eta) \leq \mathfrak{R}_n(\mathcal{F}_\eta). \tag{21}$$

Combining Lemma E.3 with standard Rademacher generalization bounds yields the derivation below.

**Theorem E.4** (Robust generalization under neighborhood smoothness). *Let $\mathcal{F}_\eta$ be the robust hypothesis class from Definition E.2, and assume $\ell$ is 1-Lipschitz and bounded in $[0, 1]$. Then for any $\delta \in (0, 1)$, with probability at least $1 - \delta$ over the draw of an i.i.d. sample $Z = \{z_t^{(1)}, \ldots, z_t^{(n)}\}$ from $\mathcal{D}$, every $f_\tau \in \mathcal{F}_\eta$ satisfies*

$$R^{\mathrm{rob}}(f_\tau) \leq \hat{R}_n^{\mathrm{rob}}(f_\tau) + 2\mathfrak{R}_n(\mathcal{F}_\eta) + 3\sqrt{\frac{\log(2/\delta)}{2n}}. \tag{22}$$

Theorem E.4 shows that neighborhood-robust training does not increase the complexity term beyond the standard Rademacher complexity of the robust function class $\mathcal{F}_\eta$. In our setting, $\mathcal{F}_\eta$ consists of student PRMs whose step-level scores are smooth with respect to the semantic neighborhoods induced by process reasoning. As a result, enforcing pointwise robustness, for example through DC-W2S's dual-consensus masking and neighborhood-aware training, controls robust generalization error without incurring an extra complexity penalty.

The robust generalization bound in Theorem E.4 captures how neighborhood smoothness controls robust risk. We now connect this to how DC-W2S selects and masks training steps, and show that focusing on P1/P3-type regions reduces an effective variance term in the bound.

# F. Proofs

Proof of Lemma E.3.

*Proof.* Fix a sample $Z = \{z_t^{(1)}, \ldots, z_t^{(n)}\}$ and Rademacher variables $\sigma = (\sigma_1, \ldots, \sigma_n)$.

We have assumed local label consistency inside neighborhoods

$$\forall z_t, \ \forall z' \in \mathrm{supp}(\mathcal{N}(z_t)) : \quad \tilde{y}_{\mathrm{agg}}(z') = \tilde{y}_{\mathrm{agg}}(z_t),$$

so that $\ell(f_\tau(z'), \tilde{y}_{\mathrm{agg}}(z')) = \ell(f_\tau(z'), \tilde{y}_{\mathrm{agg}}(z_t))$.

For brevity, write

$$\ell(f_\tau, z_t^{(i)}) \triangleq \ell\big(f_\tau(z_t^{(i)}), \tilde{y}_{\mathrm{agg}}(z_t^{(i)})\big), \qquad \tilde{\ell}(f_\tau, z_t^{(i)}) \triangleq \sup_{z' \in \mathrm{supp}(\mathcal{N}(z_t^{(i)}))} \ell\big(f_\tau(z'), \tilde{y}_{\mathrm{agg}}(z_t^{(i)})\big),$$

where we have used the above assumption of fixed supervision per base point.

Let $f_\tau \in \mathcal{F}_\eta$. By pointwise $\eta$-robustness (Definition E.2), for any $z' \in \mathrm{supp}(\mathcal{N}(z_t^{(i)}))$, $\big|f_\tau(z') - f_\tau(z_t^{(i)})\big| \leq \eta$.

Since $\ell(\cdot, \tilde{y}_{\mathrm{agg}}(z_t^{(i)}))$ is 1-Lipschitz in its first argument and bounded in $[0, 1]$, we have

$$\ell\big(f_\tau(z'), \tilde{y}_{\mathrm{agg}}(z_t^{(i)})\big) \leq \ell\big(f_\tau(z_t^{(i)}), \tilde{y}_{\mathrm{agg}}(z_t^{(i)})\big) + \big|f_\tau(z') - f_\tau(z_t^{(i)})\big| \leq \ell(f_\tau, z_t^{(i)}) + \eta.$$

Taking the supremum over $z' \in \mathcal{N}(z_t^{(i)})$ yields

$$\tilde{\ell}(f_\tau, z_t^{(i)}) = \sup_{z' \in \mathrm{supp}(\mathcal{N}(z_t^{(i)}))} \ell\big(f_\tau(z'), \tilde{y}_{\mathrm{agg}}(z_t^{(i)})\big) \leq \ell(f_\tau, z_t^{(i)}) + \eta.$$

Therefore, for any fixed $Z, \sigma$,

$$\sup_{f_\tau \in \mathcal{F}\eta} \frac{1}{n} \sum_{i=1}^{n} \sigma_i \tilde{\ell}(f_\tau, z_t^{(i)}) \leq \sup_{f_\tau \in \mathcal{F}\eta} \frac{1}{n} \sum_{i=1}^{n} \sigma_i \big(\ell(f_\tau, z_t^{(i)}) + \eta\big)$$

$$= \sup_{f_\tau \in \mathcal{F}_\eta} \frac{1}{n} \sum_{i=1}^{n} \sigma_i \ell(f_\tau, z_t^{(i)}) \; + \; \frac{\eta}{n} \sum_{i=1}^{n} \sigma_i.$$

Now take expectation over the random Rademacher variables $\sigma$ and the sample $Z$. Recall that $\mathbb{E}[\sigma_i] = 0$ for each $i$, and the $\sigma_i$ are independent of $Z$. Thus

$$\mathbb{E}_{Z,\sigma}\Big[\frac{\eta}{n} \sum_{i=1}^{n} \sigma_i\Big] = \frac{\eta}{n} \sum_{i=1}^{n} \mathbb{E}_\sigma[\sigma_i] = 0.$$

Hence

$$\mathfrak{R}_n^{\mathrm{rob}}(\mathcal{F}_\eta) = \mathbb{E}_{Z,\sigma}\left[\sup_{f_\tau \in \mathcal{F}_\eta} \frac{1}{n} \sum_{i=1}^{n} \sigma_i \tilde{\ell}(f_\tau, z_t^{(i)})\right]$$

$$\leq \mathbb{E}_{Z,\sigma}\left[\sup_{f_\tau \in \mathcal{F}_\eta} \frac{1}{n} \sum_{i=1}^{n} \sigma_i \ell(f_\tau, z_t^{(i)})\right] = \mathfrak{R}_n(\mathcal{F}_\eta).$$

This proves the lemma. □

Proof of Theorem E.4.

*Proof.* Define the robust loss class

$$\mathcal{G} \triangleq \big\{g_f : \mathcal{Z} \to [0,1] \,\big|\, g_f(z_t) = \tilde{\ell}(f_\tau, z_t), \ f_\tau \in \mathcal{F}_\eta\big\}.$$

By definition,

$$R^{\mathrm{rob}}(f) = \mathbb{E}_{z_t \sim \mathcal{D}}[g_f(z_t)], \qquad \hat{R}_n^{\mathrm{rob}}(f_\tau) = \frac{1}{n} \sum_{i=1}^{n} g_f(z_t^{(i)}).$$

Let $\mathfrak{R}_n(\mathcal{G})$ denote the (standard) empirical Rademacher complexity of $\mathcal{G}$, i.e.

$$\mathfrak{R}_n(\mathcal{G}) = \mathbb{E}_{Z,\sigma}\left[\sup_{g \in \mathcal{G}} \frac{1}{n} \sum_{i=1}^{n} \sigma_i g(z_t^{(i)})\right] = \mathfrak{R}_n^{\mathrm{rob}}(\mathcal{F}_\eta),$$

by expanding $g$ as $g_f$ and using the definition of robust complexity.

We now invoke a standard Rademacher generalization bound for bounded real-valued functions following e.g., Bartlett & Mendelson (2002).

For any class $\mathcal{G}$ of functions mapping into $[0,1]$, and for any $\delta \in (0,1)$, with probability at least $1 - \delta$ over the sample

$$\forall g \in \mathcal{G}: \quad \mathbb{E}[g(z_t)] \; \leq \; \frac{1}{n} \sum_{i=1}^{n} g(z_t^{(i)}) \; + \; 2\,\mathfrak{R}_n(\mathcal{G}) \; + \; 3\sqrt{\frac{\log(2/\delta)}{2n}}.$$

Applying this to $\mathcal{G}$ and substituting $g = g_f$, we obtain that with probability at least

$$\forall f_\tau \in \mathcal{F}_\eta: \quad R^{\mathrm{rob}}(f_\tau) \; \leq \; \hat{R}_n^{\mathrm{rob}}(f_\tau) \; + \; 2\,\mathfrak{R}_n(\mathcal{G}) \; + \; 3\sqrt{\frac{\log(2/\delta)}{2n}}.$$

Finally, noting that

$$\mathfrak{R}_n(\mathcal{G}) = \mathfrak{R}_n^{\mathrm{rob}}(\mathcal{F}_\eta) \; \leq \; \mathfrak{R}_n(\mathcal{F}_\eta)$$

by Lemma E.3.

Plugging this inequality into the bound above gives

$$\forall f_\tau \in \mathcal{F}_\eta: \quad R^{\mathrm{rob}}(f_\tau) \ \leq \ \hat{R}_n^{\mathrm{rob}}(f_\tau) \ + \ 2\,\mathfrak{R}_n(\mathcal{F}_\eta) \ + \ 3\sqrt{\frac{\log(2/\delta)}{2n}},$$

as claimed. This proves the Theorem. $\qquad\square$

