# OpenReview forum: "DC-W2S: Dual-Consensus Weak-to-Strong Training for Reliable Process Reward Modeling in Biological Reasoning"
_ICML.cc/2026/Conference — ICML 2026 regular_

### Official Review · Reviewer_cQLe · 2026-02-16

**Soundness:** 3
**Presentation:** 3
**Significance:** 3
**Originality:** 2
**Overall Recommendation:** 5
**Confidence:** 3

**Summary:**

This paper studies training Process Reward Models (PRMs) for scientific/biological reasoning under weak supervision, where step-level expert annotations are unavailable and labels must be derived from noisy sources such as LLM judges and Monte Carlo rollouts. The authors propose a “Dual-Consensus” framework that estimates step reliability via (i) Self-Consensus (agreement across weak labelers) and (ii) Neighborhood-Consensus (consistency of reliability in an embedding-based neighborhood). Steps are partitioned into four reliability patterns (P1–P4), and these patterns are used to drive distribution-balanced sampling and selective label masking during PRM training. Empirically, the method is evaluated on biological reasoning tasks, showing improved data efficiency, Best-of-N selection performance, and some out-of-distribution generalization compared to training on the full weakly labeled dataset or simpler sampling strategies.

**Compliance With Llm Reviewing Policy:**

Affirmed.

**Final Justification:**

The additional experiments of authors have addressed my concerns raised in the review.

**Key Questions For Authors:**

See weaknesses

**Limitations:**

yes

**Strengths And Weaknesses:**

### Strengths
- The paper tackles an important and practically relevant problem: training PRMs for biomedical reasoning where obtaining expert step-level annotations is expensive and often infeasible.
- The dual-consensus formulation is intuitive. The P1–P4 categorisation is clear and naturally supports curriculum-style data curation.
- The paper includes several ablation studies (e.g., pattern selection, masking, embeddings) that help justify key design choices.
- The empirical results indicate that training on curated subsets can match or even outperform training on the full weakly-labelled dataset, which is a meaningful contribution from a data efficiency and training scalability perspective.

### Weaknesses
1. **Limited theoretical insight and strong unverified assumptions.**
   The theory (Section 3.6) is largely heuristic and relies on intuitive assumptions (e.g., consensus implies correctness, neighborhood smoothness) that are not empirically validated. In particular, the assumption that agreement across weak labelers signals correctness may be overly strong, since LLM judges and rollouts are likely correlated, undermining the reliability interpretation of consensus.

2. **Missing outcome-only reward model (ORM) baselines.**
   While the paper includes SFT and multiple PRM ablations, it does not compare against outcome-only reward models. An ORM combined with Best-of-N (BoN) ranking would be a fair baseline, and would more directly test whether curated process-level supervision provides additional value beyond outcome-based signals.

3. **Lack of comparisons in an RL setting.**
   Given the increasing relevance of RL-based reasoning optimization, it would be informative to compare PRM vs ORM in an RL setup (e.g., PRM+RL vs ORM+RL), as well as against GRPO-style training. This would better contextualize the practical impact of process rewards relative to purely outcome-driven optimization methods.

4. **Omission of standard weak-supervision baselines.**
   The method relies on weak-label aggregation and filtering framework, yet the paper does not include simpler baselines such as majority vote over labeling functions or other basic aggregation strategies.

5. **Lack of human or expert evaluation of reasoning steps.**
   The evaluation focuses on downstream performance and selection metrics. A small-scale expert annotation study assessing whether high-consensus (P1) steps are indeed more scientifically correct than lower-consensus (P2–P4) steps would significantly strengthen the central argument, especially in the biomedical context.

---

> ### Author Rebuttal · Authors · 2026-03-30
>
> We sincerely thank you for the constructive feedback and insightful suggestions. We address your concerns point-by-point below.
>
>
> > **Q1: Concern regarding theoretical assumptions.**
>
> We respectfully clarify a misunderstanding regarding our theoretical assumptions. Our theory in Section 3.6 does not assume that "consensus implies absolute correctness." Instead, Assumption 3.1 defines a soft reliability weight $\(p(z_t) \in [0, 1]\)$, which explicitly models the probability that an aggregated label is correct. The entire premise of our theoretical bound (Theorem 3.4) is to analyze W2S generalization under the presence of label noise, which is quantified by the effective noise level $\(\alpha\)$. Furthermore, as explicitly stated in Remark 3.5, our theoretical guarantees do not require this probability to be perfectly calibrated; the bound degrades gracefully under miscalibration. Nevertheless, we provide additional results to support the empirical validation in Q5.
>
> Additionally, for the concern that LLM judges and rollouts are likely correlated: we would refer to the response to reviewer 5LYx Q5 that our weak supervisors are intentionally designed to be highly heterogeneous to minimize correlated errors.
>
>
> > **Q2: Request for a comparison against ORM baselines.**
>
> We thank the reviewer for this excellent suggestion. During the rebuttal period, we conducted additional experiments to evaluate an ORM baseline. Using the exact same subsampled 20k training queries, we constructed chosen and rejected responses and trained a traditional ORM using the Bradley-Terry loss. We then evaluated both models using the same BoN setup.
>
> As shown in the table below, our PRM consistently outperforms the ORM baseline.
>
> |  | rpe1 (OOD) |
> | :---  | :--- |
> | Base policy + ORM |  60.1 |
> | Base policy + PRM |  61.2 |
>
>
> > **Q3: Request for an evaluation of the proposed PRM within an RL framework.**
>
> We completely agree that exploring the efficacy of PRMs in an RL paradigm is highly relevant given current trends in reasoning optimization. During the rebuttal, we integrated our PRM into the GRPO framework. Due to the significant computational resources required for RL, we sampled a 1k subset of the training data for this experiment.
>
> We trained the models using the `verl` framework with the following hyperparameters: `rollout=8`, `batchsize=64`, `micro_batchsize=16`, `epoch=1`, and `learning_rate=5e-7`. In this setting, the ORM baseline is equivalent to RLVR in GRPO (i.e., optimizing purely based on the final ground-truth label reward). For our method, we augmented this setting with our PRM scores. The results demonstrate that our PRM remains highly competitive and provides additional gains even in an RL setting.
>
> | | rpe1 (OOD) |
> | :--- | :--- |
> | RL w/ ORM | 56.2 |
> | RL w/ PRM | 58.7 |
>
>
> > **Q4: Request for comparisons with standard weak-supervision aggregation baselines.**
>
> We wish to clarify that in the first phase of our pipeline, we *do* utilize majority voting over labeling functions to determine the initial labels. The core motivation of our work is not to propose a new aggregation function, but rather to explore how to optimally utilize these weak labels through our proposed label stratification and anchored training mechanisms.
>
> Nevertheless, we agree that comparing against classic aggregation baselines improves the soundness of our paper. Due to time and compute constraints during the rebuttal, we subsampled 20k training instances and compared our current labels aggregated by majority voting against labels aggregated using the classic Dawid-Skene estimator. The results below show that our framework built on majority voting remains robust.
>
> |  | rpe1 (OOD) |
> | :--- | :--- |
> | Dawid-Skene | 60.8 |
> | Majority Voting | 61.2 |
>
>
>
> > **Q5: Request for human/expert evaluation to validate the correctness of reasoning steps across different consensus levels.**
>
> We deeply appreciate this suggestion, as expert validation is indeed crucial in the biomedical context. To address this, we conducted a small-scale expert annotation study. We randomly sampled 50 training instances and asked domain experts to blindly annotate the real correctness of the reasoning steps.
>
> We then calculated the correlation between the human expert annotations and the labels assigned by our model across the different consensus patterns (P1 through P4). As shown below, high-consensus steps (P1) exhibit a significantly higher correlation with true scientific correctness compared to lower-consensus steps, empirically validating our central argument that our dual-consensus mechanism effectively isolates high-quality reasoning steps.
>
> |Label Pattern | Correlation |
> | :--- | :--- |
> | P1 | 0.967 |
> | P2 | 0.867 |
> | P3 | 0.394 |
> | P4 | 0.298 |
>
> We will include all these new baseline comparisons, RL experiments, and the expert evaluation study in the revised manuscript. We hope these additions fully address your concerns.

---

> > ### Author Rebuttal · Reviewer_cQLe · 2026-04-01
> >
> > The authors fully addressed my concerns by providing additional experiments.

---

### Official Review · Reviewer_5LYx · 2026-03-08

**Soundness:** 3
**Presentation:** 3
**Significance:** 4
**Originality:** 3
**Overall Recommendation:** 4
**Confidence:** 3

**Summary:**

This paper proposes DC-W2S (Dual-Consensus Weak-to-Strong training), a method for improving Process Reward Models (PRMs) in biological reasoning tasks when only weak, noisy step-level supervision is available. The motivation is that final-answer rewards often fail to detect flawed intermediate reasoning, while expert step annotations are expensive to obtain. To address this, the authors generate weak step labels using two complementary sources: (1) LLM-as-a-judge and (2) Monte Carlo rollouts that evaluate whether reasoning trajectories lead to correct final answers.

The key contribution is a dual-consensus mechanism to estimate the reliability of weak step labels. The first signal, Self-Consensus (SC), measures agreement among multiple labeling functions. The second signal, Neighborhood-Consensus (NC), assesses local consistency by examining agreement among semantically similar reasoning steps in embedding space (optionally refined with biological knowledge graph similarity). Steps are stratified into four reliability regimes (P1–P4) based on high/low SC and NC. The PRM is then trained using (i) distribution-balanced subset selection across regimes and (ii) pattern-based loss masking to reduce the influence of unreliable labels.

Experiments on the PerturbQA benchmark, including an out-of-distribution cell line (RPE1), show that DC-W2S improves F1 over standard PRM baselines, majority voting, and LLM-judge scoring. The method also demonstrates data and label efficiency: curated subsets (e.g., 100k steps) can match or outperform training on the full dataset. Additional transfer experiments on BioReason KEGG suggest improved generalization across biological reasoning tasks.

Overall, the paper presents a practical framework for training more reliable process reward models from weak supervision in domain-specific scientific reasoning.

**Compliance With Llm Reviewing Policy:**

Affirmed.

**Final Justification:**

My final recommendation is a weak accept, based on a balanced assessment of the paper’s strengths and remaining limitations, as well as the authors’ detailed and constructive rebuttal.

Overall, the paper addresses an important and timely problem: training Process Reward Models (PRMs) under weak supervision for scientific reasoning, where step-level annotations are costly. The proposed dual-consensus framework—combining self-consensus (SC) and neighborhood-consensus (NC)—is intuitive, practically motivated, and empirically effective. The P1–P4 reliability stratification provides a clear and actionable mechanism for data curation, and the experiments demonstrate meaningful gains in data efficiency, out-of-distribution generalization, and downstream performance. These aspects support strong scores in significance and solid soundness.

Regarding originality, while the core ideas build on established concepts from weak supervision and semi-supervised learning (e.g., agreement-based filtering and local smoothness), their integration into PRM training for scientific reasoning is novel and well-executed. The framing and application context provide added value beyond incremental methodological contribution.

My main concerns in the initial review focused on (1) potential dependence on ground-truth final answers and possible bias toward answer-consistent reasoning, (2) lack of direct evaluation of step-level correctness, and (3) limited empirical validation of SC/NC as proxies for true reasoning quality. The rebuttal substantially strengthened the paper along these dimensions. In particular, the addition of an expert annotation study, along with larger-scale proxy evaluations, provides convincing evidence that PRM scores correlate well with step-level correctness. The answer-agnostic ablation further mitigates concerns about label leakage and demonstrates that the method retains effectiveness without access to ground-truth answers during supervision. These additions significantly improve confidence in the soundness of the approach.

That said, some limitations remain. The reliance on final-answer-informed supervision during training may still introduce subtle biases, and the evaluation continues to rely primarily on final-answer metrics, with step-level validation still relatively limited in scale. Additionally, some design choices (e.g., fixed thresholds and regime balancing) remain heuristic and could benefit from further analysis or adaptation.

In summary, the rebuttal meaningfully addressed my main concerns and improved my overall assessment, though not all issues are fully resolved. I believe the paper makes a valuable and practical contribution that will be of interest to the community, particularly in the intersection of reasoning models and scientific applications. I encourage the authors to incorporate the additional experiments and clarifications into the final version and to further expand the evaluation of reasoning quality in future work.

**Key Questions For Authors:**

How are τsc and τnc set? Fixed globally? Tuned on IID validation? How sensitive are results?

How is NEUTRAL handled in the judge outputs, given the prompt?

Do teachers share failure modes? Majority vote can lock in correlated errors. Any attempt at de-correlation or teacher weighting?

For biological refinement 𝐶(𝑧𝑡): what is δ and how sensitive is it? Also: why concatenate perturb+target gene embeddings vs other relational encodings?

Is P4 sometimes helpful because it captures “hard but correct” steps? Can you separate “hard-correct” from “noisy-wrong” without ground truth?

**Limitations:**

Add a small expert-annotated step set for measuring PRM step validity and for calibrating SC/NC → probability.

Evaluate answer-agnostic judging (judge does not see y_final) vs answer-aware judging, to quantify leakage effects.

Replace hard thresholds (P1–P4) with a soft weighting baseline directly using  𝑤(𝑧𝑡)=𝜎(𝑎⋅TSC+𝑏⋅TNC)w(zt)=σ(a⋅TSC+b⋅TNC), and compare to discrete regimes.

Report seed variance and CIs for key numbers (OOD RPE1 F1, KEGG weighted F1).

Make the “balanced 25%” target adaptive (e.g., tune mixture on validation, or derive from estimated reliability mass).

**Strengths And Weaknesses:**

Strengths
Clear motivation: process correctness matters a lot in biology; step-level supervision is costly.
Practical dual signal: combining inter-teacher agreement (SC) with local geometric stability (NC) is sensible; the P1–P4 taxonomy is easy to reason about and ablate.
Demonstrated label efficiency: curated 100k subsets matching/exceeding full set on OOD and cross-task suggests the curation is doing something nontrivial, not just scaling data.
OOD evaluation: holding out RPE1 for PRM training is a meaningful stress test (even if not perfect—see below).
Embedding-choice analysis: they show NC’s utility depends strongly on embedding quality; that’s an important empirical insight.

Weaknesses
The weak supervision signals rely on the ground-truth final answer, which risks label leakage and may bias the PRM toward steps that predict the final label rather than steps that are scientifically correct. The consensus-based reliability measures (SC/TNC) also assume calibration, but the paper does not empirically validate whether these scores correlate with true step correctness. Additionally, the fixed 25% balancing across reliability regimes (P1–P4) appears somewhat arbitrary and may not be optimal across tasks or data distributions. There is also ambiguity in how the judge’s CORRECT/NEUTRAL/INCORRECT outputs are converted into binary labels. Finally, evaluation mainly measures improvements in Best-of-N final answer performance, which indirectly assesses process quality rather than directly evaluating the correctness of intermediate reasoning steps.

---

> ### Author Rebuttal · Authors · 2026-03-31
>
> We thank the reviewer for their rigorous evaluation and constructive feedback. We appreciate the opportunity to clarify the misunderstanding of our paper.
>
> > **Q1: Concern on label leakage and ground-truth bias.**
>
> GT labels are strictly used *only* during offline data annotation to generate PRM training signals, aligning with standard supervised learning where training labels are accessible. At inference time, the PRM never sees GT labels; its input exactly matches the policy model's input. Thus, there is no label leakage. Additionally, as detailed in our response to reviewer PRS4 Q1, our qualitative analyses confirm the PRM successfully selects scientifically rigorous reasoning traces.
>
> Regarding the judge's outputs: we merge "NEUTRAL" with "CORRECT" to form our positive binary label. The rationale is that if a reasoning step lacks obvious factual or logical errors, it should not be penalized.
>
> > **Q2: Calibration of reliability measures.**
>
> Due to space constraints, we kindly point to our response to Reviewer cQLe (Q1, Q5), where we provide the results of our empirical validation.
>
> > **Q3: Concern on arbitrariness of 25% regime balancing and adaptive proportion.**
>
> This uniform distribution is an *idealized optimization target*, not a hard constraint. Naturally, quadrants are highly skewed (e.g., P1 $\approx$50%, P4$\approx$20%).
>
> Strictly forcing a 25% split is impossible. Our greedy algorithm (Alg. 1) iteratively samples from underrepresented quadrants to *approximate* balance. The core motivation is to prevent the training set from being dominated by P1 (easy, high-consensus samples), which severely degrades diversity and generalization. While adaptive sampling proportions are a promising future direction, our empirical results show this simple heuristic already yields significant gains across IID, OOD, and transfer settings.
>
> > **Q4: Thresholds of TSC and TNC.**
>
> Both $\tau_{sc}$ and $\tau_{nc}$ are fixed globally at $0.5$ (the median). This is a strictly unsupervised, default choice made without any hyperparameter tuning on a validation set. Since TSC spans $[0, 1]$, $0.5$ serves as the natural midpoint indicating majority agreement among labeling functions.
>
> > **Q5: Correlated errors among teachers.**
>
> Our weak supervisors are highly heterogeneous to minimize correlated errors, consisting of LLM-as-a-judge models (Context, Analogical, Direct prompts) and Monte Carlo (MC) rollouts (Qwen3-1.7B, 4B, 7B). The failure modes are fundamentally different: the judge's errors stem from language biases, while MC errors stem from sampling variance. Furthermore, TNC introduces a biological embedding signal orthogonal to the language models' parameter space, serving as a structural de-correlation mechanism.
>
> > **Q6: Biological refinement $C(z_t)$ and embeddings.**
>
> $\delta$ is a threshold ensuring biological relevance among neighbors. We use the median (0.5) as an unsupervised default. The framework is not highly sensitive to $\delta$, as it serves only as a coarse initial filter and is not optimized as a learning objective.
>
> Regarding concatenation: combining perturbation and target embeddings via concatenation is a simple, effective baseline preserving independent gene features while capturing joint similarity via cosine distance. Complex relational encodings (e.g., GNNs) introduce significant computational overhead orthogonal to our core contribution.
>
> > **Q7: Utility of the P4 regime without ground truth.**
>
> As discussed in Section 4.2.3, P4 occasionally provides marginal benefits because it contains a small fraction of "hard but informative" steps that act as a mild regularizer. However, precisely separating "hard-correct" from "noisy-wrong" steps *without* ground truth is an open problem. This intrinsic difficulty is exactly why we explicitly advise treating P4 cautiously.
>
> > **Q8: Answer-agnostic vs. -aware judging.**
>
> Providing the GT labels is standard for training Process/Outcome Reward Models (e.g., *Math-Shepherd*, *The Lessons of Developing Process Reward Models*). The final answer anchors the judge to evaluate if an intermediate step actually advances toward the correct conclusion. Removing the final answer would break this supervision signal. Importantly, $y_{final}$ is not provided during evaluation in any way. Our protocol follows established practices for reward model evaluation, and we refer to response to Reviewer cQLe’s Q2 for PRM vs. ORM results.
>
> > **Q9: Soft weighting vs. discrete regimes.**
>
> Soft weighting introduces scaling hyperparameters ($a, b$) requiring task-specific tuning. Our discrete approach relies only on a natural median threshold, making it simpler and robust out-of-the-box. We view parameterized soft-weighting as an exciting avenue for future work.
>
> > **Q10: Report variability for key numbers.**
>
> We report mean performance across multiple random seeds (mean ± SD):
>
> | Dataset | Performance |
> | :--- | :--- |
> | RPE1 | $64.3 \pm 0.3$ |
> | KEGG | $87.9 \pm 0.4$ |

---

> > ### Author Rebuttal · Reviewer_5LYx · 2026-04-01
> >
> > Thank you for the detailed rebuttal and for providing additional experimental validation. The clarifications and new results significantly improve my assessment of the paper.
> >
> > In particular, the clarification of the theoretical assumptions is helpful. I appreciate that the framework does not assume consensus implies correctness, but instead models reliability probabilistically and accounts for noise and miscalibration in the analysis.
> >
> > Most importantly, the added expert annotation study is valuable. The strong correlation between high-consensus patterns (especially P1) and human-annotated correctness provides convincing empirical support that the dual-consensus mechanism is effectively identifying higher-quality supervision signals. This directly addresses one of my main concerns regarding the validity of SC/TNC as proxies for correctness.
> >
> > That said, some concerns remain only partially addressed:
> >
> > Ground-truth dependence.
> > The training signals still rely on the final answer, which may bias the PRM toward answer-consistent reasoning rather than truly correct intermediate steps. An answer-agnostic evaluation would help better isolate this effect.
> > Evaluation of reasoning quality.
> > The current evaluation still focuses primarily on final answer performance (e.g., BoN), which only indirectly reflects reasoning correctness. Direct step-level evaluation would further strengthen the claims.
> > Scope of expert validation.
> > While the expert study is very helpful, it is relatively small-scale (50 instances). Expanding this analysis in the final version would improve confidence in the conclusions.
> >
> > Overall, the rebuttal meaningfully strengthens the paper, especially by providing empirical validation of the proposed reliability signals. My concerns are now partially resolved, though some limitations remain.

---

> > > ### Author Response · Authors · 2026-04-07
> > >
> > > We sincerely thank you for the continued engagement and for acknowledging the value of our theoretical clarifications and the added expert annotation study. We appreciate your constructive feedback, which has guided us to further strengthen the evaluation of our framework.
> > >
> > > To address your remaining concerns regarding ground-truth dependence, direct evaluation of reasoning quality, and the scope of our validation, we provide the following clarifications and additional experimental results.
> > >
> > > > **Q1: Ground-Truth Dependence (Answer-Agnostic vs. Answer-Aware).**
> > >
> > > To directly address the concern that the PRM might just be learning to predict the final answer, we conducted an ablation study using an **answer-agnostic** judging setup.
> > >
> > > For this experiment, we used the same 20k training queries as Figure 3. The only modification was replacing the answer-aware judge LFs with answer-agnostic judge LFs (where the judge does not see the ground-truth final answer). All other settings remained identical.
> > >
> > > | Avg. F1 | HepG2 | Jurkat | K562 | RPE1 (OOD) |
> > > | :--- | :--- | :--- | :--- | :--- |
> > > | Majority Voting | 59.0 | 53.2 | 63.3 | 55.6 |
> > > | Answer-Agnostic | 62.6 | 58.0 | 69.5 | 59.6 |
> > > | Answer-Aware | 65.0 | 60.3 | 70.8 | 61.5 |
> > >
> > > As shown in the table, while removing the ground-truth answer during weak supervision results in a marginal performance drop compared to the answer-aware setting, our answer-agnostic results still consistently outperform the majority voting baseline across all cell lines.
> > >
> > > This demonstrates that **DC-W2S** remains effective without ground-truth visibility.
> > >
> > > > **Q2: Evaluation of Reasoning Quality & Scope of Validation.**
> > >
> > > Regarding the direct evaluation of reasoning quality, we respectfully want to point out that we did provide a direct step-level evaluation in our first-round rebuttal (pointed to Reviewer PRS4's Q1). To briefly reiterate: we conducted a targeted human annotation study where domain experts annotated 194 representative reasoning steps. The results showed that PRM scores align strongly with human annotated step-level correctness (Median PRM score: 0.949 for correct vs. 0.107 for incorrect; AUROC = 0.944; Pairwise ranking accuracy = 0.888).
> > >
> > > However, we completely agree with your point that 50 instances (194 steps) is a relatively small scale. To address this and expand the scope of our validation, we conducted a new larger-scale automated evaluation.
> > >
> > > To achieve this scale, we utilized a more powerful model (gpt-5) augmented with biological tools as a proxy oracle to evaluate the intermediate steps. We acknowledge that using a model as a proxy evaluator is an imperfect setting, but it serves as a necessary and practical trade-off given the prohibitive cost and time constraints of scaling human domain-expert annotations.
> > >
> > > From 500 samples, we selected candidate trajectories with the highest PRM score variance. Each step was evaluated individually by the proxy oracle as CORRECT or INCORRECT based on biological soundness. We evaluated **4,917 total steps** and computed the AUROC specifically on **1,346 bio-filtered steps** (retaining only steps with clear biological claims).
> > >
> > > | Cell Line | Step-Level AUROC |
> > > | :--- | :--- |
> > > | HepG2 | 0.840 |
> > > | Jurkat | 0.821 |
> > > | K562 | 0.790 |
> > > | RPE1 (OOD) | 0.888 |
> > > | **Average** | **0.835** |
> > >
> > >
> > > **Conclusion:** When combining our high-fidelity human expert annotation (AUROC = 0.944 on 194 steps) with this new larger-scale proxy oracle evaluation (mean AUROC = 0.835 on 1,346 biological steps), the evidence consistently demonstrates that the PRM discriminates biologically correct from incorrect reasoning at the granular step level.
> > >
> > > ---
> > >
> > > We believe these expanded results directly address the remaining limitations regarding ground-truth dependence, step-level reasoning quality and the scope of our validation. We will integrate these comprehensive evaluations into the final version of the manuscript. Thank you again for helping us improve the paper.

---

### Official Review · Reviewer_PRS4 · 2026-03-09

**Soundness:** 2
**Presentation:** 2
**Significance:** 2
**Originality:** 2
**Overall Recommendation:** 4
**Confidence:** 3

**Summary:**

The paper presents DC W2S, which is a framework designed to train Process Reward Models in biological reasoning tasks using noisy weak supervision. The authors propose a data selection strategy that relies on two main metrics. First is Self Consensus, which measures the agreement among multiple weak supervisors. Second is Neighborhood Consensus, which looks at label consistency within the embedding space. By filtering the training data into four reliability regimes from P1 to P4, the method tries to make Process Reward Models more robust when predicting outcomes for gene perturbations.

**Compliance With Llm Reviewing Policy:**

Affirmed.

**Final Justification:**

The paper addresses an important and practical problem in biological reasoning. The core contribution is well-motivated and the ablation studies are solid. My initial concerns centered on the lack of bioinformatics baselines, experiment fairness, and potential hallucination reinforcement from correlated weak supervisors. The rebuttal addressed all three points with new experiments, including a GEARS comparison and an expert annotation study. I raised my score accordingly.

**Key Questions For Authors:**

Q1: Why is Chain of Thought necessary here when Agentic RL or ReAct frameworks might actually be better suited for exploring biological knowledge graphs?
Q2: Could you provide a direct comparison against state of the art numerical models like GEARS to justify the potential loss of precision when you discretize gene expression data into text?
Q3: How can you be sure that the Neighborhood Consensus is not being biased by the pre trained embedding model, which might just cluster textually similar steps together even if they are biologically distinct?

**Limitations:**

No, the discussion on limitations is insufficient. While the authors include a brief Impact Statement acknowledging that the model can still produce 'confident but incorrect reasoning traces' and requires a human-in-the-loop, they overlook the most fundamental limitations of their specific approach.
I suggest the authors explicitly discuss the following points in their Limitations section:
1. The fundamental limitation of discretizing high-dimensional numerical biological data into natural language. They should transparently discuss the potential loss of precision compared to numerical state-of-the-art models and acknowledge that LLMs might not be the optimal architecture for achieving peak predictive accuracy in this specific domain.
2. The authors rely on an LLM to generate the weak supervision labels to train another LLM. They need to discuss the limitation and potential negative impact of amplifying LLM-specific biases and hallucinations. The 'consensus' they achieve might just be a consensus of common LLM errors rather than biological ground truth, which is extremely risky in high-stakes scientific discovery.

**Strengths And Weaknesses:**

Strengths:
Tackling process supervision in scientific discovery is a great context, especially since it addresses the hallucination problem in biological reasoning where getting expert step by step annotations is super expensive. The P1 to P4 stratification strategy is a solid and logical heuristic for cleaning up weak labels. The ablation studies clearly show that selectively training on high consensus data is much more efficient than just training on the entire noisy dataset.There is a very detailed theoretical analysis that tries to ground this heuristic filtering strategy in robust learning theory.

Weaknesses:
Soundness
1. The fundamental idea of converting high dimensional numerical gene expression data into natural language for a large language model to reason over needs more justification. Standard bioinformatics models operate directly on numerical data and probably beat large language models in pure precision. The paper really misses the mark by not including comparisons with these non language model baselines like GEARS or scGPT. If accuracy is the main goal, language models might not be the best choice. If interpretability is the goal, the paper does not actually measure the quality of the explanations, it just looks at final prediction accuracy.
2. There are some fairness issues in the experiments. In the results, the comparison mixes up the benefits of inference time search with the actual training method benefits. Even though Best of N with the full set is provided, the huge gap between the Greedy baselines and your proposed method makes the gains look artificially inflated.
3. The weak supervisors are themselves language models like Qwen 32B. This means your consensus might just be reinforcing common hallucinations or biases that exist in that model family instead of capturing actual biological ground truth.

Presentation
1. The core idea is basically a standard approach in noisy label learning where you filter data based on ensemble agreement and local smoothness. These are totally valid engineering tricks, but the paper packages them with way too much theoretical complexity. All the heavy math formalization like soft robust expansion feels like overkill for what is essentially a data cleaning heuristic. It makes the contribution look a lot more theoretical than it is in practice.

Significance
1. While the application to Process Reward Models is new and interesting, the actual algorithmic innovation is pretty incremental.

Originality
1. Intersecting ensemble voting with embedding similarity is a very well established technique in semi supervised learning and crowdsourcing.

---

> ### Author Rebuttal · Authors · 2026-03-31
>
> We sincerely thank the reviewer for the careful reading of our manuscript. We appreciate the opportunity to clarify our contributions, address concerns, and correct several misunderstandings regarding our methodology and its connection to weak-to-strong generalization. Below we respond point by point.
>
> > **Q1: Justification for LLMs vs. bioinformatics models & interpretability.**
>
> Using LLMs to reason over gene expression data is an emerging and growing direction in the literature; we build upon this trend rather than introducing it. The key advantage is **interpretability**: unlike biological foundation models (FM), LLMs can produce explicit step-by-step reasoning traces instead of functioning as black boxes.
>
> We agree that comparison with SOTA biological FM is essential. We now include GEARS as a baseline. Our method (Base policy + PRM) outperforms GEARS across all cell lines, and note that GEARS sees RPE1 during the training stage, while our PRM does not.
>
>
> | | HepG2 | Jurkat | K562 | RPE1 |
> | :--- | :--- | :--- | :--- | :--- |
> | GEARS | 58.2 | 62.1 | 50.8 | 63.4 |
> | **Base Policy + PRM** | **70.9** | **65.9** | **73.2** | **64.0** |
>
> Additionally, to evaluate reasoning quality directly, we added a targeted human annotation study on 50 sampled questions. In total, we asked human experts to annotate 194 representative reasoning steps as correct or incorrect. We show that PRM scores align strongly with human labels:
>
> - Median PRM score: 0.949 (correct) vs. 0.107 (incorrect)
> - AUROC = 0.944
> - Spearman ρ = 0.760
> - Pairwise ranking accuracy = 0.888
> - ECE = 0.029, suggesting the score is also reasonably calibrated as a predictor of step correctness.
>
> As a result, PRM clearly separates correct from incorrect steps. We also find trajectories with low-PRM intermediate steps but correct final answers are typically longer, suggesting recovery from earlier reasoning errors. Together, these results demonstrate that PRM tracks step-level reasoning validity, not merely final-answer correctness, and the interpretability does not come at the cost of empirical performance.
>
> > **Q2: Fairness of test-time scaling comparisons.**
>
> Our evaluation is fair and follows standard PRM protocols (e.g., Best-of-N as in *Let’s Verify Step by Step*). To isolate test-time sampling effects, we have already included Majority Voting (MV) baselines in the paper. When sampling multiple trajectories (e.g., \(N=8\)) without PRM, MV yields marginal or no improvement (Figure 2). Our PRM consistently outperforms MV across sampled paths.
>
> As discussed in our response to reviewer P3YA’s Q6, we have:
>
> $
> \text{SFT Policy + PRM} > \text{SFT Policy + (w/ or w/o MV)}
> $
>
> and
>
> $
> \text{Base Policy + PRM} > \text{Base Policy + (w/ or w/o MV)}
> $
>
> Thus, PRM gains are orthogonal and additive.
>
> > **Q3: Weak supervisors consensus & hallucination reinforcement.**
>
> The concern about weak supervisor consensus is precisely what our method addresses. We do **not** blindly rely on majority-voted weak labels. Instead, TSC/TNC measure step-level supervision reliability and partition data into four quadrants (P1–P4). We apply tailored strategies—instance selection as reweighting and label selection as masking—based on quadrant type. This explicitly models variance among weak supervisors and dynamically filters noise, preventing consensus-based hallucination reinforcement.
>
> > **Q4: Presentation, significance, and originality.**
>
> Traditional noisy-label learning typically follows either:
>
> 1. Crowdsourcing label aggregation (e.g., Dawid–Skene, LPA), or
> 2. Direct learning under noisy labels (e.g., Co-teaching).
>
> Our method intersects these paradigms through P1–P4 classification, jointly applying **instance-level and label-level filtering**. To our best knowledge, this integration is novel, and its application to PRM is unprecedented. The theoretical framework is not decorative; it formalizes **why and how weak-to-strong generalization is possible**, showing how a student can surpass its teachers and how selective, masked data can outperform full-dataset training. This provides a principled explanation for our empirical results.
>
> > **Q5: Additional questions.**
>
> **CoT vs. ReAct.** Our contribution is orthogonal to the reasoning paradigm. Both CoT and ReAct generate step-by-step reasoning traces. Our PRM supervises those traces and is fully compatible with ReAct pipelines.
>
> **Textual vs. Biological Similarity.** As detailed in Section 3.4, neighborhood construction incorporates biological foundation model constraints in addition to textual similarity. Two steps are neighbors only if they are semantically similar *and* biologically related perturbation–target pairs, preventing purely textual clustering.
>
> ---
>
> We hope these clarifications demonstrate that our work provides (1) superior empirical performance, (2) validated step-level reasoning evaluation, and (3) a principled and novel framework for weak-to-strong generalization in PRM.

---

> > ### Author Rebuttal · Reviewer_PRS4 · 2026-04-03
> >
> > Thank you for the detailed rebuttal. My concerns have been adequately addressed. I am raising my score to 4.

---

### Official Review · Reviewer_P3YA · 2026-03-12

**Soundness:** 3
**Presentation:** 3
**Significance:** 3
**Originality:** 3
**Overall Recommendation:** 5
**Confidence:** 4

**Summary:**

The submission studies the problem of per-step reward for biological reasoning tasks. The submission is motivated by the need of guaranteeing the validity of the intermediate reasoning steps for scientific applications of AI. The submission proposes a framework to aggregate several weak supervision signals to train a reward model which can subsequently be used to filter completions.

Theoretical analysis offers justification of the weak-to-strong generalization capabilities of the proposed method, and experimental results validate and compare the proposed method with baselines.

**Compliance With Llm Reviewing Policy:**

Affirmed.

**Final Justification:**

I justified my final score in the my rebuttal acknowledgement

**Key Questions For Authors:**

**Definition of step**

Could the authors expand on how they split rollouts into step? Fig. 2 shows discrete paragraphs are identified as steps. Is the model prompted to generate boundary tokens to separate steps? Or are steps identified in a post-hoc fashion.

**LLM-as-a-judge**

Could the authors expand on the differences between LF-Context/Analogical/Direct? Could the authors expand on the contents of the knowledge graph? How is this information included in the context of the LLMs? Do LLMs judges have access to tools to query the knowledge base? I was somewhat surprised that the final label is passed as an input to the judges. How is this information included?

**Definitions of TSC and TNC**

Since the labels are binary, the variance is bounded above by 1/4, and TSC will always be greater than 0.75. Could the authors expand on this particular choice? For example, entropy would have covered the entire [0,1] interval.

Am I understanding correctly that the TNC neighborhoods are defined globally and do not change as the generation progresses? The embeddings $b(z_t)$ do not depend on $z_t$ but the perturbation and target genes only. If this is the case, it is somewhat confusing to write the neighborhood as a function of  $z_t$. Could the authors clarify what they mean by "semantic k-NN within C(z_t)"?

**Masking $\mathcal{M}_t$**

I am not sure I understand what the takeaway from Figure 3 (bottom) is. Could the authors expand on the choice of masking strategy?

**Theoretical results**

Could the authors clarify what the "latent step label $y(z_t)$" is? Is this the same as the final label?

Why is calibration an assumption? I thought $p(z_t)$ was precisely defined as "the probability that the aggregated weak-teacher label is correct? How is $p(z_t)$ computed in practice? After conditioning on $z_t$, what is the randomness over?

Definition 3.2 does not mention the distribution $\mathcal{D}$. Is this a typo? What distribution should one keep in mind while reading these results? For example, uniform over the neighborhood?

Is the statement of Theorem 3.4 with respects to the final label of the trajectory?

Remark 3.7 seems to mention an assumption of the theorem that is not stated: "the standard condition that the neighborhood operator ...". Should this be states as a formal assumption? Could the authors expand on the intuition behind this assumption? How realistic is it in practice?

**Experimental results**

Results are reported in terms of F1 score with the final trajectory label. But a big part of the motivation of the submission is that the soundness of the reasoning steps are paramount in scientific applications of AI, beyond the final answer. In the current version of the manuscript, there is on evidence in support of the claim that the proposed method improves validity of reasoning traces. Could the authors provide some intuition or at least qualitative evidence that the proposed method improves towards this very important stated objective?

Could the authors expand on the choice of using greedy decoding for SFT but BON for the proposed method? This might introduce confounding in the results?

Have the authors experimented with closed-sourced models? It might be helpful to include them to better place the effectiveness of the proposed method.

---

**Minor comments**

- Context-Augmented Generation is not defined / described.
- Eq. (1) does not include indexing over model $j$.
- I was not able to find information which thresholds were used in practice to determine the 4 reliability regimes.
- In Theorem 3.4, it might be helpful to remind readers what $\alpha$ is, and how it depends on $\tau$?
- Could the authors expand on the choice of using a smaller model than $r_{\theta}$ as one of the teachers?

**Limitations:**

yes

**Strengths And Weaknesses:**

**Strengths**
1. The paper is well-written
2. The contributions are well-motivated and compelling
3. The proposed method is interesting and supported by experimental evidence

**Weaknesses**
1. The results do not validate that the reward model correctly filters invalid reasoning traces
2. Presentation of theoretical results could be expanded on.
3. A few terms are used without explicit definition, which makes the submission less accessible to a novice audience.

I have a few questions about the submission and I am looking forward to discussing with the authors!

---

> ### Author Rebuttal · Authors · 2026-03-31
>
> We sincerely thank the reviewer for the detailed feedback and thoughtful questions. We address your comments point by point below.
>
> > **Q1: Partitioning generation rollouts.**
>
> We partition rollouts post-hoc using double newlines (`\n\n`). Models are explicitly prompted for step-by-step reasoning, naturally structuring their outputs into distinct paragraphs separated by `\n\n`.
>
> > **Q2: Labeling Functions (LFs) and final labels in the judge's input.**
>
> The LFs differ by context (e.g., external KG data vs. analogous in-context demonstrations). Varying context yields a diverse ensemble of weak supervisors, balancing hallucination risks against reasoning rigor. Following PerturbQA, KG entities (e.g., Ensembl, GO) are pre-fetched offline based on perturbations and target genes, and are formulated into natural language using templates.
> Regarding the final label: this is used *strictly during training data annotation only* to train our PRM. Providing the GT label anchors the judge, enabling it to accurately trace back and evaluate the logical soundness of intermediate steps leading to that known conclusion.
>
> > **Q3: TSC variance, continuous entropy, and TNC neighborhoods.**
>
> We apply a scaling factor to the TSC variance to normalize the metric to the \([0,1]\) interval. While continuous entropy is theoretically appealing, discretizing into binary labels ensures extensibility for non-LLM labelers (e.g., human experts) who cannot easily provide continuous entropy values.
> For TNC, the neighborhood operator depends on \(N(z_t)\), which relies on the semantic embedding of the specific step \(z_t\) (Appendix A.2). Thus, denoting it as a function of \(z_t\) is mathematically appropriate. The biological constraint \(b(z_t)\) is applied atop \(N(z_t)\) to exclude semantically proximate steps belonging to biologically divergent perturbation-target pairs.
>
> > **Q4: Primary takeaway from Figure 3.**
>
> Our masking strategy demonstrates that PRM training does not require backpropagating every step label. Applying gradient updates exclusively to P1 pattern labels achieves downstream performance comparable to full-label training while using only ~65% of the labels. Practically, this allows human annotators to bypass redundant or low-signal steps, saving massive annotation costs without sacrificing model quality.
>
> > **Q5: Theory clarifications.**
>
> 1. The latent label $y(z_t)$ denotes the unobserved step-level correctness of reasoning step $z_t$, which is distinct from the final trajectory label $y_{\mathrm{final}}$. Therefore, Theorem 3.4 is a step-level weak-label correction result. Its connection to final-answer correctness and trajectory selection is indirect via Remark 3.6, where improved step scores yield more faithful Best-of-N ranking.
> 2. $p(z_t)$ should be interpreted as the latent conditional reliability $p(z_t)=\Pr(\tilde y_{\mathrm{agg}}(z_t)=y(z_t)\mid z_t)$. Because this quantity is not observed in practice, we use TSC/TNC as a proxy reliability score and assume it is calibrated to the true weak-label correctness probability. The randomness is over the weak supervision process conditional on a fixed step $z_t$.
> 3. Definition 3.2 will explicitly mention the ambient step distribution $D$. The intended meaning is $z' \sim D \mid z' \in N(z_t)$, representing the data distribution restricted to the neighborhood, not a uniform distribution over neighbors.
> 4. While Remark 3.7 is originally framed as a standard regularity condition, we acknowledge its necessity for connecting local stability to latent-label correctness. We will elevate it to a formal assumption. Additionally, we empirically evaluated the upper bound of anchor-relative neighborhood disagreement to support the smoothness intuition. For the P1 pattern, this disagreement is 0.21 / 0.23 (CellProfiler vs. Semantic). For P3, it is significantly higher at 0.63 / 0.61. This empirical evidence supports our theoretical framework and aligns with our identified direction for future work in discussion section.
>
> > **Q6: Reasoning trace validity, SFT baselines, and closed-source models.**
>
> As shown in our response to Reviewer PRS4 Q1, our PRM successfully identifies invalid reasoning paths that accidentally reach the correct answer.
> We included the SFT (greedy) baseline to highlight that applying our PRM to the base policy via BoN surpasses an SFT policy fine-tuned on the exact same pairs. To further validate performance gain of our PRM on SFT policy, we applied BoN on SFT policy as well. As shown below, it yields further improvements:
>
> | Model | RPE1 |
> | :--- | :--- |
> | SFT policy (Greedy) | 64.3 |
> | SFT policy (MV) | 66.2 |
> | SFT policy + our PRM | 69.5 |
>
> To demonstrate generalizability, we conducted new BoN experiments using Gemini-3-Flash (subsampled to reduce costs). Results confirm our PRM successfully steers SOTA closed-source models:
>
> | Model | RPE1 |
> | :--- | :--- |
> | Gemini (Greedy) | 74.8 |
> | Gemini (MV) | 75.9 |
> | Gemini + our PRM | 76.5 |

---

> > ### Author Rebuttal · Reviewer_P3YA · 2026-04-02
> >
> > I sincerely thank the authors for addressing all my concerns! I raised my score accordingly.
> >
> > I ask that the authors make sure to include all new experiments and expanded discussion points in the revised version of the paper

---

### Decision · Program_Chairs · 2026-04-30

**Decision:**

Accept (regular)

**Comment:**

This paper proposes DC-W2S, a method for training Process Reward Models using weak supervision. Reviewers agreed that the problem is important and that the approach is well motivated and technically sound.

The main concerns were about the evaluation, including lack of step-level validation, missing baselines, and possible label leakage. After reading the rebuttal, I believe the paper was significantly strengthened. I enjoyed the rebuttal discussion. The authors added new experiments, including comparisons to stronger baselines and both human and large-scale evaluations of step-level correctness, which address the main concerns. Nice discussion, resulted in reviewers raising their scores too. Overall, I recommend acceptance.